# Metagenomic profiles of archaea and bacteria within thermal and geochemical gradients of the Guaymas Basin deep subsurface

Paraskevi Mara [1,5], David Geller-McGrath[2,5], Virginia Edgcomb [1], David Beaudoin[1], Yuki Morono [3] & Andreas Teske [4] ✉

Previous studies of microbial communities in subseafloor sediments reported that microbial abundance and diversity decrease with sediment depth and age, and microbes dominating at depth tend to be a subset of the local seafloor community. However, the existence of geographically widespread, subsurface-adapted specialists is also possible. Here, we use metagenomic and metatranscriptomic analyses of the hydrothermally heated, sediment layers of Guaymas Basin (Gulf of California, Mexico) to examine the distribution and activity patterns of bacteria and archaea along thermal, geochemical and cell count gradients. We find that the composition and distribution of metagenome-assembled genomes (MAGs), dominated by numerous lineages of Chloroflexota and Thermoproteota, correlate with biogeochemical parameters as long as temperatures remain moderate, but downcore increasing temperatures beyond ca. 45 °C override other factors. Consistently, MAG size and diversity decrease with increasing temperature, indicating a downcore winnowing of the subsurface biosphere. By contrast, specific archaeal MAGs within the Thermoproteota and Hadarchaeota increase in relative abundance and in recruitment of transcriptome reads towards deeper, hotter sediments, marking the transition towards a specialized deep, hot biosphere.

The interplay between temperature stress and energy availability determines microbial survival in the subsurface biosphere, and delineates the extent and limits of life in the deep subsurface biosphere[1,2]. As microbial communities in cool, relatively shallow subsurface sediments transition into more deeply buried and increasingly warm and finally hot sediments, it should be possible to track how subsurface bacteria and archaea react to these gradually harsher regimes downcore on the levels of cellular activity and community change. While microbial abundance and diversity are generally expected to decline downcore[3,4], it is also possible that particular subsurface-adapted microbial populations benefit from conditions that would eliminate others, and constitute a specialized deep, hot biosphere. Recent studies indicated active microbial populations in extremely deep and hot sediments, yet without sequence-based identification[5,6]. To learn more about bacterial and archaeal communities of the deep, hot biosphere from a genomic perspective,

[1]Geology and Geophysics Department, Woods Hole Oceanographic Institution, Woods Hole, MA 02543, USA. [2]Biology Department, Woods Hole Oceanographic Institution, Woods Hole, MA 02543, USA. [3]Kochi Institute for Core Sample Research, Institute for Extra-cutting-edge Science and Technology Avantgarde Research (X-STAR), Japan Agency for Marine-Earth Science and Technology (JAMSTEC), Monobe, Nankoku, Kochi, Japan. [4]Department of Earth, Marine and Environmental Sciences, University of North Carolina at Chapel Hill, Chapel Hill, NC 27599, USA. [5]These authors contributed equally: Paraskevi Mara, David Geller-McGrath. ✉e-mail: teske@email.unc.edu

downcore trends of diversity and activity in increasingly hot sediments need to be examined, and microbial communities and their genomes have to be tracked downcore, as far as microbial biomass and DNA yield allow. Yet, investigating downcore changes in microbial abundance, community composition and activity in well-characterized geochemical and thermal gradients requires a suitable field site where extensive physical, chemical and microbial gradients can be sampled in adequate resolution by sediment coring and drilling.

An ideal natural laboratory for such a research task is Guaymas Basin, a hydrothermally-active ocean spreading center in the Gulf of California, covered by several hundred meters of sediment that host basaltic sill intrusions[7] and strong geothermal heat flow[8]. Pyrolysis of buried organic carbon in these organic-rich sediments produces a complex milieu of petroleum hydrocarbons, including light hydrocarbons and methane, alkanes, and aromatic compounds, as well as carboxylic acids, and ammonia[9,10]. These compounds are transported via hydrothermal fluids through Guaymas Basin's thick sediments, supporting diverse and active microbial communities[11]. Collectively, these communities not only perform chemosynthetic carbon fixation and heterotrophic organic matter remineralization, but they also assimilate fossil carbon into the benthic biosphere[12]. Yet, few studies to date have explored the microbiology of deep subsurface sediments in Guaymas Basin. Methanogens were enriched from sediments collected during Deep Sea Drilling Program Expedition 64 to Guaymas Basin[13], and bacterial and archaeal communities in piston cores were surveyed using 16S rRNA amplicon sequencing[14–16]. Aside from these studies, the spatial extent, diversity and activity of the deep biosphere in Guaymas Basin have remained largely unknown.

International Ocean Discovery Program (IODP) Expedition 385 drilled into Guaymas Basin at eight locations that differ in their degree of hydrothermal influence and heatflow[8], to survey their resulting characteristics[17]. Drilling sites followed broadly a northwest-to-southeast transect across the northern Guaymas axial trough (Fig. 1). Two neighboring sites (U1545 and U1546) on the northwestern end of

Guaymas Basin[18,19] essentially differ by the presence of a massive, thermally equilibrated sill between 350 to 430 meters below seafloor (mbsf) at Site U1546[7]. Two drilling sites (U1547, U1548) targeted the hydrothermally active Ringvent area, approximately 28 km northwest of the spreading center[15], where a shallow, recently emplaced and hot sill creates steep thermal gradients and drives hydrothermal circulation[20]. Drilling Site U1549[21] explores the periphery of an off-axis methane cold seep, Octopus Mound, located ~9.5 km northwest of the northern axial graben[22].

These contrasting sites provide an opportunity for a comprehensive analysis of subsurface microbiota at different temperatures and depths. To assess the environmental distribution and genomic potential of microbes living in the deep biosphere of Guaymas Basin, we analyzed reconstructed metagenome-assembled genomes (MAGs) from depths ranging from 0.8 to 219.5 mbsf at these thermally and geochemically contrasting sites. We also provide evidence for the activity of specific bacterial and archaeal lineages by mRNA transcript mapping on bacterial and archaeal MAGs.

## Results and discussion
### Sampling sites and depths
Metagenomes were produced from sediment samples at drilling sites U1545B to U1549B that follow a northwest-to-southeast transect across the northwestern flanking region of Guaymas Basin (Fig. 1A) and include an off-axis hydrothermal system, the Ringvent site (Fig. 1B). The samples were selected to coordinate with depths used for separate ongoing analyses, and ranged from 1.7 m to 219.5 mbsf at Site U1545B, 0.8-16.3 mbsf at U1546B, 2.1-75.7 mbsf at U1547B, 9.1-69.4 mbsf at U1548B, and 16.5 mbsf at U1549B (Fig. 1; Table 1). For all samples, a wide range of geochemical parameters was analyzed shipboard (Supplementary Dataset 1). The sites represent distinctly different thermal gradients and cell densities; generally, sites with steeper downcore temperature gradients are characterized by more rapidly decreasing cell counts (Fig. 1C, D). U1545B is the reference site for IODP Expedition

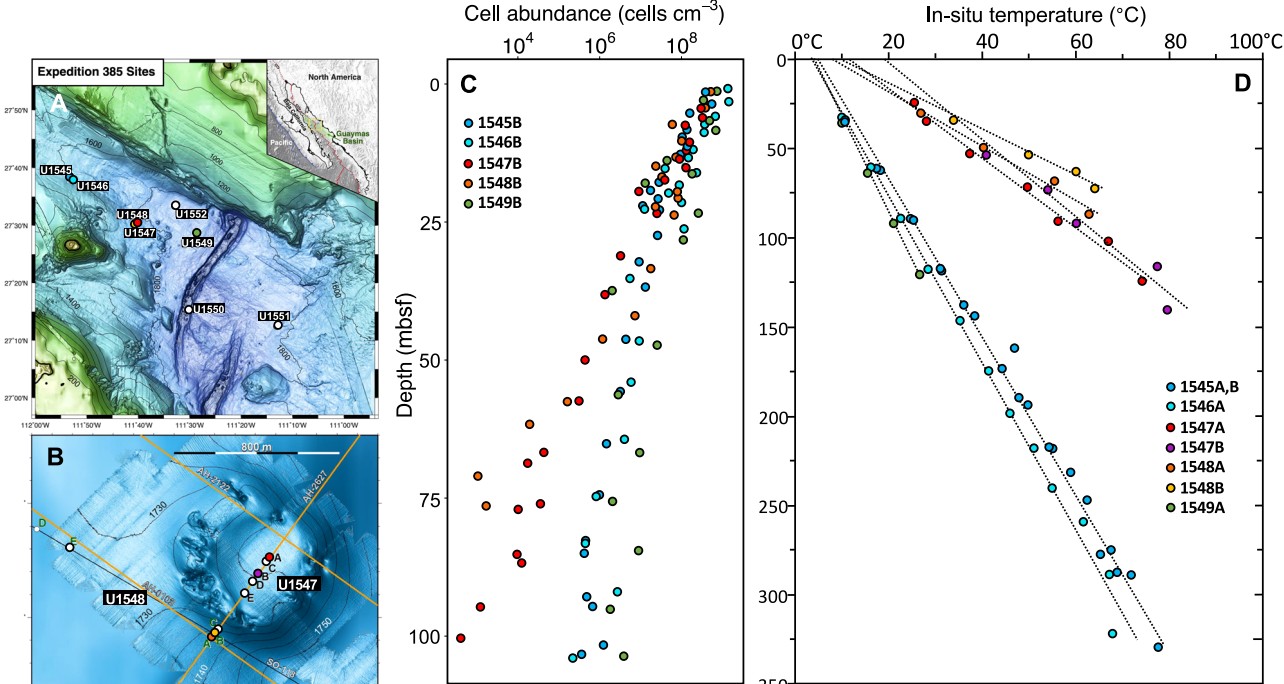

**Fig. 1 | Locations, cell count profiles and temperature profiles for IODP Expedition 385 drilling sites. A** Guaymas Basin bathymetry with drill sites. **B** Bathymetry of Ringvent with drill sites within and on the periphery of the Ringvent site. **C** Cell counts for drill sites (U1545, U1546, U1547, U1548, and U1549) where metagenomic and metatranscriptomic samples were collected. **D** Temperature

profiles for drill sites where metagenomic and metatranscriptomic samples were collected. The lines indicated linear functions that were fitted to in-situ temperature measurements. Bathymetric maps, courtesy of D. Lizarralde (WHOI). Cell count and temperature data are provided in the Source Data file.

**Table 1 | Geochemical, depth and temperature data for metagenomic samples**

| | Sample ID | T°C | Depth (mbsf) | Alkalinity (mM) | SO₄²⁻ (mM) | PO₄³⁻ (µM) | H₂S (µM) | NH₄⁺ (mM) | CH₄ (mM) | CO (nM) | DOC (mg/L) | DIC (mM) | TOC (wt%) | TN (wt%) | TOC/TN |
|---|---|---|---|---|---|---|---|---|---|---|---|---|---|---|---|
| **U1545B** | U1545B_1H2 | 5.3 | 1.7 | 6 | 26.9 | 33.9 | 44.1 | 0.5 | 0 | 360 | 24.2 | 2.4 | 4.87 | 0.61 | 9.3 |
| | U1545B_2H3 | 6.4 | 6.8 | 6.9 | 26.3 | 36.4 | 1220.5 | 0.5 | 0 | 367 | 24.2 | 2.4 | 4.03 | 0.63 | 7.45 |
| | U1545B_4H2 | 10.4 | 24.3 | 21 | 21.1 | 51 | 6068 | 5.3 | 0 | 189 | 25.7 | 7.6 | 2.43 | 0.38 | 7.4 |
| | U1545B_4H3 | 10.7 | 25.8 | 21 | 21.1 | 51 | 6068 | 5.3 | 0 | 254 | 25.7 | 7.6 | 2.43 | 0.38 | 7.4 |
| | U1545B_8H3 | 19.3 | 63.8 | 59.5 | 0.7 | 78.3 | 8947 | 9.2 | 1.5 | 174 | 73.2 | 27.9 | 2.73 | 0.34 | 9.4 |
| | U1545B_13H4 | 30.2 | 112.5 | 40.4 | 0.4 | 77.1 | 1891 | 15.1 | 1.2 | 161 | 52.3 | 14.1 | 1.97 | 0.29 | 8.0 |
| | U1545B_19F3 | 39.8 | 155.0 | 35.3 | 0.3 | 46.9 | 3.2 | 15.9 | 1.9 | 116 | 48.2 | 10.7 | 2.17 | 0.34 | 7.5 |
| | U1545B_32F3 | 52.4 | 211.1 | 28.5 | 0.3 | 16.2 | 0 | 23.9 | 0.4 | 160 | 50.7 | 4.8 | 1.74 | 0.28 | 7.3 |
| | U1545B_34F3 | 54.3 | 219.5 | 26.7 | 0.2 | 12.2 | 0 | 25.6 | 0.6 | 168 | 40.9 | 3 | 2.42 | 0.4 | 7 |
| **U1546B** | U1546B_1H2 | 2.8 | 0.8 | 5.1 | 27.7 | 12.4 | 428 | 0.7 | 0 | 665 | 21.8 | 2.2 | 4.36 | 0.52 | 9.7 |
| | U1546B_3H2 | 6.2 | 16.4 | 7.3 | 26.7 | 28.7 | 1226.6 | 0.3 | 0 | 260 | 21 | 2.5 | 2.15 | 0.34 | 7.4 |
| **U1547B** | U1547B_1H2 | 14.2 | 2.2 | 3 | 27.9 | 16.5 | 68 | 0.1 | 0 | 350 | 13.9 | 1.2 | 3.76 | 0.45 | 9.7 |
| | U1547B_1H3 | 15.0 | 3.6 | 3 | 27.9 | 16.5 | 68 | 0.1 | 0 | 350 | 13.9 | 1.2 | 3.76 | 0.45 | 9.7 |
| | U1547B_2H2 | 17.5 | 8.7 | 5.3 | 26.8 | 22.8 | 759 | 0.3 | 0 | 427 | 14.7 | 1.2 | 3.53 | 0.29 | 14 |
| | U1547B_2H3 | 17.8 | 9.9 | 5.3 | 26.8 | 22.8 | 759 | 0.3 | 0 | 427 | 14.7 | 1.2 | 3.53 | 0.29 | 14 |
| | U1547B_3H3 | 23.7 | 19.3 | 7.8 | 26.1 | 33.3 | 1538 | 0.6 | 0 | 252 | 14.6 | 1.5 | 2.13 | 0.26 | 9.4 |
| | U1547B_5H2 | 31.9 | 36.9 | 14.5 | 23.3 | 24.7 | 3725.5 | 1.5 | 0 | 197 | 14.3 | 1.8 | 2.63 | 0.29 | 10.5 |
| | U1547B_5H3 | 34.0 | 38.1 | 14.5 | 23.3 | 24.7 | 3725.5 | 1.5 | 0 | 197 | 14.3 | 1.8 | 2.63 | 0.29 | 10.5 |
| | U1547B_7H3 | 42.3 | 57.4 | 13.3 | 21.1 | 27 | 5110 | 2.6 | 0 | 83 | 23.3 | 1.5 | 2.03 | 0.26 | 9 |
| | U1547B_8H2 | 46.6 | 65.8 | 13 | 20.2 | 25.4 | 6606 | 2.9 | 0 | 91 | 23.3 | 1.5 | 1.25 | 0.15 | 9.5 |
| | U1547B_9H2 | 51.0 | 74.3 | 13.7 | 18.8 | 28 | 7151 | 3.7 | 0 | 54 | 15 | 1.9 | 2.2 | 0.26 | 10 |
| | U1547B_9H3 | 51.8 | 76.0 | 13.7 | 18.8 | 28 | 7151 | 3.7 | 0 | 54 | 15 | 1.9 | 2.2 | 0.26 | 10 |
| **U1548B** | U1548B_2H3 | 13.7 | 8.9 | 4.8 | 26.8 | 24.5 | 732.5 | 0.3 | 0 | 554 | 13.1 | 1.4 | 2.4 | 0.22 | 12.9 |
| | U1548B_4H7 | 33.5 | 33.5 | 9.5 | 25.4 | 18.6 | 1913 | 0.6 | 0 | 303 | 9.4 | 1.2 | 1.9 | 0.22 | 10.3 |
| | U1548B_8H5 | 62.4 | 69.5 | 13 | 20.2 | 25.4 | 6606 | 2.9 | 0 | 51 | 1.5 | 0.2 | 1.57 | 0.2 | 9 |
| **U1549B** | U1549B_3H2 | 6.4 | 16.5 | 18.7 | 17.5 | 98.1 | 2567 | 3.3 | 0 | 188 | 23.6 | 5.7 | 3.12 | 0.59 | 6.1 |

The samples are sorted by increasing temperature. "Sample ID" is composed by site and core section IDs. Geochemical data are compiled from Expedition 385 site chapters using the best available sample matches, and in situ temperatures represent linear interpolation based on published temperature gradients in the site chapters[18–21].

385 because of the absence of seepage, hydrothermal influence, and massive sill intrusions[18]. Here, metagenome libraries extended down to 219.5 mbsf, at in-situ temperatures of 54.3 °C. Cell count trends for sites U1545, U1546 and U1549 were similar, and showed a decrease over three orders of magnitude within 100 meters (Fig. 1C). At the hot Ringvent sites U1547B and U1548B[8,20], comparable temperatures of 50–55 °C were already reached near 70 mbsf (Fig. 1D), and cell counts decreased by four to five orders of magnitude within this depth range (Fig. 1C). To describe temperature-related trends in MAG recovery and diversity, we categorized our samples into three groups according to temperature; cool (2–20 °C), warm (20–45 °C) and hot (>45 °C).

**Subsurface Biogeochemical zonation**

Most samples collected for metagenomes are from sediments within the sulfate-reducing zone where sulfate is still available at near-seawater concentrations (~28 mM) or becomes gradually depleted with depth (Table 1). At those same sediments hydrogen sulfide concentrations are gradually increasing towards multiple millimolar concentrations. Metagenome samples from site U1545B also include depths spanning the sulfate-methane transition zone (SMTZ) at ~64 mbsf where sulfate is consumed by microbial sulfate reduction, and methane begins to accumulate. At the SMTZ sulfate concentrations drop from 21.1 mM to 0.7 mM, sulfide reaches peak concentrations of 8.9 mM, and methane concentrations increase from picomolar to 1.5 mM (Table 1). High methane concentrations persist also in deeper samples from U1545B, and decrease only in the very deepest samples (> 200 msbf). The deep subsurface methane reservoir at this and other sites results from long-term thermogenic and biological methane accumulation[23]. In contrast to site U1545B, samples from Ringvent sites

U1547B and U1548B show gradual downcore sulfate consumption (from 27.9 to 18.8 mM) but not depletion, combined with hydrogen sulfide accumulation (max. 7.1 mM at 75.7 mbsf at U1547B); methane does not accumulate in these samples. Ammonia concentrations increase from < 1 mM towards 3 to 5 mM downcore at most sites, and reach 9 to 25 mM below the SMTZ in U1545B. Dissolved inorganic carbon (DIC) and alkalinity concentrations are generally highest at Site U1545B where they peak in the SMTZ (~28 and 60 mM, respectively). Ammonia, DIC and alkalinity remain elevated not only in the upper sediment column but also in the deeper samples of Site U1545B, presumably due to cumulative bioremineralization of buried organic matter over time at this undisturbed site. In contrast, the Ringvent samples (sites U1547B and U1548B) generally have lower ammonia, alkalinity and DIC porewater concentrations, suggesting reduced remineralization of organic matter at these sites, most likely a consequence of hydrothermal activity due to recent volcanic sill emplacement[15]. Dissolved organic carbon (DOC) remained ~10 to 20 mg/L in most samples but increased towards 70 mg/L in the sulfate-methane transition zone of U1545B and remained between 20 and 50 mg/L in the deeper sediments of U1545B. This suggests DOC enrichment and decreased heterotrophic DOC consumption in deep methanogenic sediments of U1545B where energy-rich electron acceptors for heterotrophic carbon remineralization are not available. While total nitrogen and total organic carbon generally decrease with depth at all sites, the Ringvent sites have moderately elevated TOC/TN ratios (Table 1), likely reflecting the influence of nitrogen-depleted hydrothermal carbon sources[14]. Total petroleum hydrocarbon, saturated and polyaromatic hydrocarbon content remain each quite similar across a wide range of sediments and temperatures, before

increasing considerably in hot sediments (>80 °C) near deep sill intrusions (Supplementary Dataset 1, and Supplementary Figs. 1A, B).

## MAG diversity, distribution, and evidence of activity

A total of 142 metagenome-assembled genomes (MAGs) were recovered from a co-assembly of all metagenomic samples (Supplementary Dataset 2). MAGs that matched those from negative controls were excluded from further analysis (Supplementary Dataset 3). For downstream analysis, we retained 89 bacterial and archaeal MAGs that had at least ≥ 50% bin completeness and ≤ 10% bin contamination[24]. Genome completeness ranged from ~50 to 97% (Supplementary Fig. 2; Supplementary Dataset 4).

Of these 89 MAGs, 26 MAGs were assigned to 6 archaeal phyla, and 63 MAGs were assigned to 13 bacterial phyla (Fig. 2); the phylogenetic spectrum includes lineages documented previously in 16S rRNA gene amplicon sequencing of shallow subsurface sediments[14,15], and in metagenomic surveys of shallow hydrothermal sediments of Guaymas Basin[25]. In parallel to downcore decreasing cell numbers (Fig. 1), MAG diversity decreased downcore at all sites as temperatures increased (Fig. 2). In samples from cool (2–20 °C) sediments from all sites, reads mapped to diverse bacterial and archaeal phyla, including the bacterial phyla Chloroflexota, Acidobacteriota, Desulfobacterota, WOR-3, Aerophobota, and Bipolaricaulota, and the archaeal phyla Thermoproteota, Thermoplasmatota, and Aenigmatarchaeota (Fig. 2). In samples with warm temperatures (20–45 °C), reads were predominantly assigned to bacterial phyla Chloroflexota (mostly order-level group G1F9), Acidobacteriota, WOR-3 (order-level group UBA3073), Aerophobota and Bipolaricaulota, and to archaeal phyla Thermoproteota, Hadarchaeota, and Aenigmatarchaeota. At hot temperatures (45–60 °C), bacterial reads mapped primarily to a single Chloroflexota MAG (class Dehalococcoidia), a single WOR-3 MAG and two Aerophobota MAGs (class Aerophobia). In contrast, several Archaeal MAGs show a marked preference for hot sediments, and mapped to the Thermoproteota (class Bathyarchaeia), and Hadarchaeota (class Hadarchaeia). Our recovered MAGs reflected metabolisms predicted for the deep biosphere including sulfur, nitrogen and methane cycling, hydrocarbon degradation, and carbon fixation[26] (Supplementary Note, and Supplementary Figs. 3 and 4). Desulfobacterota MAGs linked to sulfate reduction contained the *dsr* operon (e.g., *dsrB/J/K/D*) that is essential for dissimilatory sulfate reduction[27], and were recovered from shallow sediments with available sulfate. These MAGs also shared potential for iron reduction using extracellular electron transfer mechanisms, such as *mtrA*, *mtoA*, and *eetB* genes[28,29]. Marker genes of dissimilatory iron reduction[28] (e.g., *dmkA*, *dmkB*, *eetA*, *eetB*, *fmnA*, *fmnB*, *pplA*, *ndh2*) with the potential for extracellular electron transfer (EET) were identified in 86/89 MAGs within all recovered phyla (Supplementary Note, and Supplementary Datasets 5, 6).

To determine any intra-phylum differences in metabolic activity, we mapped reads of the Guaymas Basin subsurface metatranscriptome[30] to our recovered MAGs, for samples collected at the same sites (Fig. 3). Since the metagenome and metatranscriptome of the Guaymas Basin subsurface remain incompletely covered by sequence data, the absence of transcript read mapping to particular MAGs cannot be taken as evidence of microbial inactivity. Microbial activity of the deep biosphere is certainly constrained but not eliminated by substrate and energy limitation[1]. To avoid these ambiguities that are inherent in negative transcript mapping results, we focus on positive transcript mapping results that support the activity of specific MAGs in the subsurface. Actively transcribed genes are present for MAGs within all phyla discussed here, albeit at variable levels; some MAGs within individual phyla show no or much lower apparent activity than others (Fig. 3).

Most transcriptionally active bacterial and archaeal MAGs from warm and hot sediments belong to uncultured lineages previously detected in hydrothermal chimneys, sulfidic springs and seeps, and in Guaymas Basin surficial hydrothermal sediments (Fig. 3). Bacterial transcripts from warm sediments were affiliated with four MAGs (GMP_018, GMP_083, GMP_036, and GMP_057) of the Chloroflexota GIF19 lineage, a dominant group in carbonate hydrothermal chimneys[31]. Other transcripts from warm sediments mapped to MAG GMP_019 within the dehalogenating Dehalococcoides lineage, to MAGs GMP_007 and GMP_011 within the subsurface Aerophobota, and MAG GMP_58 within the Bipolaricaudota lineage UBA7950, found at the Lost City hydrothermal vents[32]. Archaeal transcripts in warm and hot sediment samples were mapped to MAG GMP_008 within the Hadarchaeota, MAG GMP_075 of the Aenigmatarchaeota QMZP01 lineage from a terrestrial sulfur spring[33], MAG GMP_040l within the thermoproteotal brine pool lineage TCS64[34], and to three Thermoproteota MAGs GMP_002, GMP_026, and GMP_039 within the B26-1 lineage from Guaymas Basin hydrothermal sediments[35]. Transcriptional activity of these MAGs suggests their inherent physiological adaptations to warm and reducing habitats are advantageous in the Guaymas Basin subsurface as well.

## The influence of environmental factors on MAG composition

The relationship between environmental parameters (Supplementary Dataset 1) and the taxonomic composition of MAGs from cool (2–20 °C), warm (20–45 °C), and hot temperatures (45–62 °C) was investigated using non-metric multidimensional scaling (nMDS) (Fig. 4) and Canonical Correspondence Analysis (CCA) (Supplementary Fig. 5). In both analyses, bacterial and archaeal MAGs clustered consistently by temperature, comparable to previous analyses of temperature-dependent microbial community composition in surficial sediments in Guaymas Basin[22]. In particular, MAGs from hot sediments aligned with temperature as the strongest influencing factor (Fig. 4). Total sulfide concentration ($H_2S$) was aligned with temperature in the CCA plot (Supplementary Fig. 5). For samples from warm sediments, nMDS analyses revealed that methane, alkalinity, and dissolved inorganic carbon (DIC) concentrations exerted a significant effect ($p < 0.05$) on the MAG community (Fig. 4). For cool sediments, nMDS and CCA showed consistently that MAGs clustered in the direction of total organic carbon (TOC) and total nitrogen (TN) content, CO concentration, pH and salinity (Fig. 4, Supplementary Fig. 5). The influence of TN and TOC on MAG diversity in cool samples may reflect increased availability of labile sources of dissolved and particulate organic matter in near-surface sediments. The consistent impact of pH and salinity on MAG diversity in cool samples, in both CCA and nMDS analyses, reflects persistent downcore trends towards lower pH and slightly reduced porewater salinity (Supplementary Dataset 1).

To summarize, the environmental parameters that impact MAG composition change downcore, from surface-linked factors such as TN, TOC, pH and salinity that impact MAGs in cool sediments, to biogeochemical parameters reflecting terminal organic matter degradation, such as increasing DIC, alkalinity and methane concentrations, in deeper and warmer sediments. An organic substrate-depleted, DIC- and methane-enriched deep subsurface environment may select for specific phyla or taxa with autotrophic capabilities (e.g., Hadarchaeota). For MAGs from deep and hot samples, carbon or nitrogen substrates, or other chemical factors, become secondary to the impact of temperature itself.

## Metagenomic features with wide subsurface distribution

In addition to genes for core metabolic processes (e.g., glycolysis, biosynthesis of nucleotides and amino acids), Guaymas Basin MAGs contain widespread genomic features that extend across multiple bacterial and archaeal phyla. Some of these widespread genomic features have obvious adaptive value and are thus retained for survival, while others challenge assumptions on subsurface adaptations and evolutionary constraints under subsurface conditions.

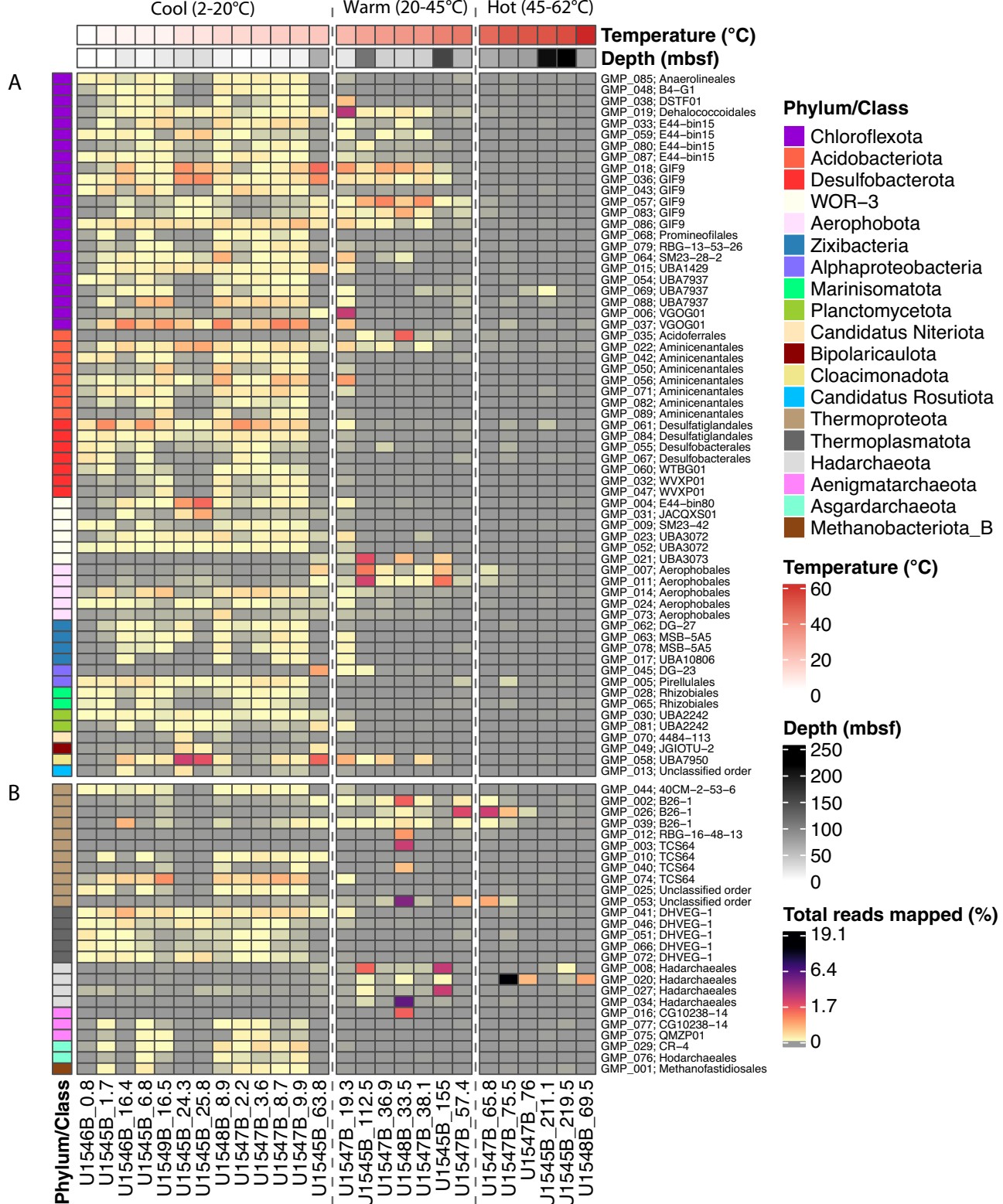

**Fig. 2 | Heatmap of MAG relative abundance.** Each column shows the percentage of total pre-processed metagenomic reads (relative abundance) that mapped to all 89 MAGs, for samples ordered by increasing temperature from left to right on the x-axis (annotated by site numbers and depths in mbsf). Temperature regimes (Cool, Warm, and Hot) are separated by vertical dashed lines. Each row shows the abundance profile of an individual MAG across all samples. MAGs are color-coded by phylum on the left, and annotated by GMP (Guaymas MAG Prokaryote) numbers 001 to 089 and order-level affinity to the right. RKPM, reads mapped per kilobase of genome, per million mapped reads. Panel section **A** denotes bacterial MAGs and panel section **B** denotes archaeal MAGs. Relative MAG abundances are provided in the Source Data file.

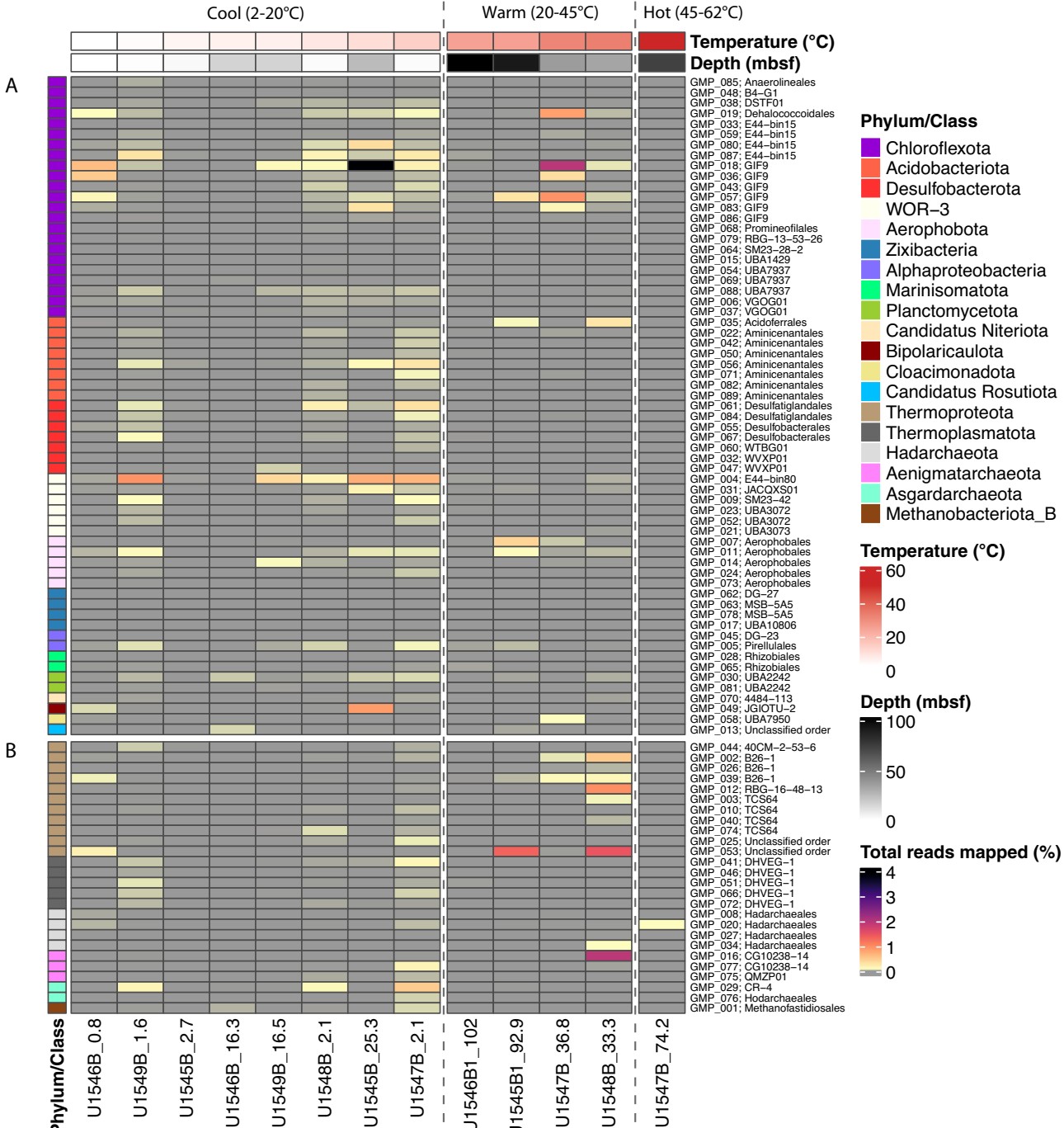

**Fig. 3 | Heatmap of MAG Metatranscriptomic read recruitment.** Each column shows the percentage of total pre-processed metatranscriptome reads (relative abundance) that mapped to all 89 MAGs, for samples ordered by increasing temperature from left to right on the x-axis (annotated by site numbers and depths in mbsf). Temperature regimes (Cool, Warm, and Hot) are separated by vertical dashed lines. Each row shows the abundance profile of an individual MAG across all samples. MAGs are color-coded by phylum on the left, and annotated by GMP (Guaymas MAG Prokaryote) numbers 001 to 089 and order-level affinity to the right. RKPM, reads mapped per kilobase of genome, per million mapped reads. Panel section **A** denotes bacterial MAGs and panel section **B** denotes archaeal MAGs. Relative transcript abundances are provided in the Source Data file.

Among genes that confer survival advantages, two-component systems (TCSs) can induce metabolic shifts, and are used extensively by bacteria and some archaea to respond and adapt to environmental changes[36]. Generally, archaea acquire TCS genes through horizontal gene transfer from bacteria[37]. In Guaymas Basin, TCS genes occur in the majority of bacterial MAGs but not in archaeal MAGs (Supplementary Datasets 5, 6), and they may help

cells to adapt to long term burial. For example, the KinABCDE-Spo0FA system is present in almost all our bacterial MAGs and plays a role in sporulation by shifting cellular metabolism from active growth to dormancy/sporulation[38]. Likewise, the RegB/RegA redox-signaling mechanism involved in carbon fixation, hydrogen oxidation and anaerobic respiration[39] is present in the majority of the bacterial MAGs.

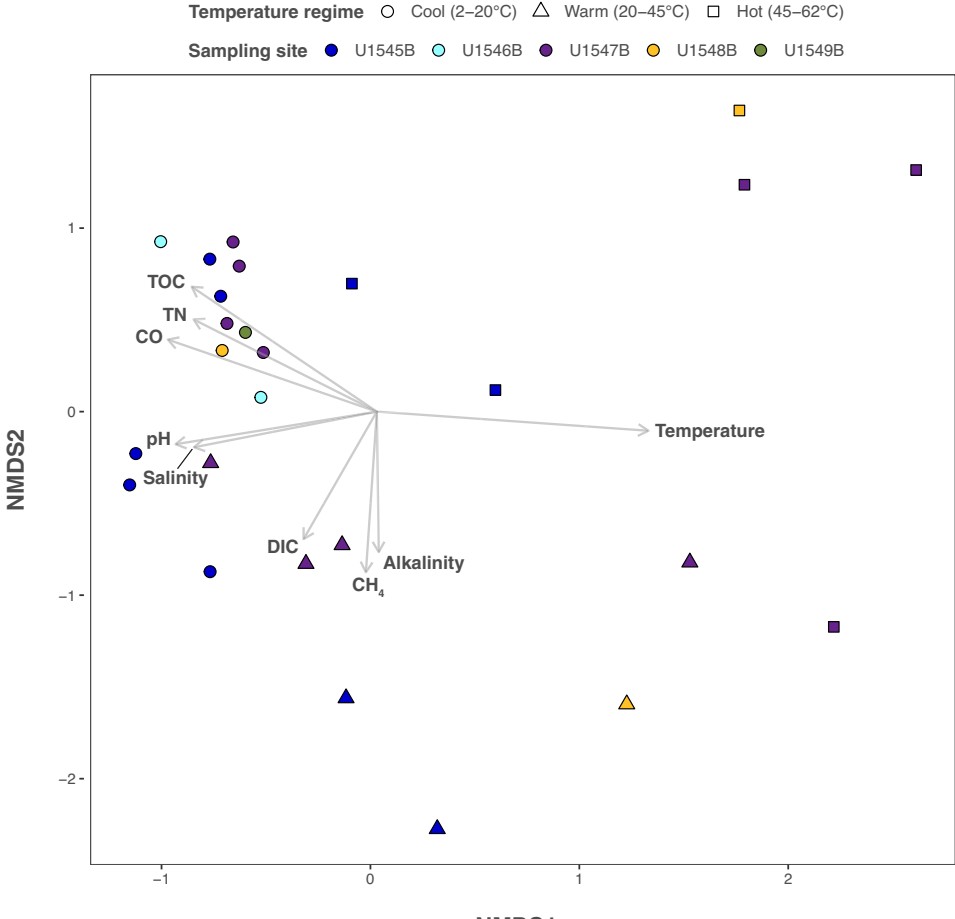

**Fig. 4 | Non-metric multidimensional scaling (nMDS) ordination plot of MAGs and environmental parameters.** The nMDS plot depicts the correlation of Guaymas Basin MAG occurrence with in-situ environmental parameters (plot stress: 0.106). On the basis of Fisher's method for combining $p$-values, we show environmental variables with $p$-values < 0.05 resulting from a two-sided permutation test. The directions of the arrows indicate a positive or negative correlation among the environmental parameters with the ordination axes (temperature, $p = 0.0001$; pH, $p = 0.0041$; salinity, $p = 0.0119$; alkalinity, $p = 0.0445$; dissolved inorganic carbon (DIC), $p = 0.0457$; methane ($CH_4$), $p = 0.0215$; carbon monoxide (CO), $p = 0.0011$; total organic carbon (TOC), $p = 0.0003$; total nitrogen (TN), $p = 0.0024$). Arrow length reflects correlation strength between environmental parameter and MAG occurrence. The samples are color-coded by site, and their temperature regimes are indicated by shape (circles for 2–20 °C; triangles for 20–45 °C and squares for 45–62 °C). nMDS plot parameters for samples and their MAGs are provided in the Source Data file.

Among widely distributed genes, we found a variety of transporters and efflux pumps associated with microbial defense, and biosynthetic gene clusters involved in synthesis of diverse secondary metabolites (Supplementary Datasets 5, 6). While all bacterial and archaeal MAGs encoded transporters[26], efflux pumps (found in 67% of all MAGs) included a large proportion of multidrug resistance pumps, detected in, 44% of all MAGs. In 25% of all MAGs, biosynthetic gene clusters were involved in the biosynthesis of diverse secondary metabolites (Supplementary Dataset 7). Archaeal biosynthetic gene clusters were primarily annotated as polyketide synthases, ribosomally synthesized and post-translationally modified peptides. Additionally, archaeal genes encoded the synthesis of terpenes (e.g., geranylgeranyl diphosphate synthase; Supplementary Dataset 6) that can be part of their lipid membranes, or function as pigments, antimicrobial agents and (in plants) as thermoprotectants[40].

Genes involved in chemotaxis (*cheA/B/R/W/Y*) and motility (*flgB/C/ E/G/H/I* and *fliE/F/G/*) were present in 56% of all bacterial and archaeal MAGs (Supplementary Datasets 5, 6). These findings suggest potential for cell-cell interaction, cell movement and competition for resources in the Guaymas Basin subsurface microbial community – a surprising result given deep biosphere microorganisms are trapped in tight pore spaces that limit movement and interaction[41]. While cell motility genes are gradually depleted downcore in the marine subsurface[42], they do not disappear. Cell motility and secondary metabolite biosynthesis genes were present and expressed in marine subsurface MAGs from Peru Margin and Canterbury Basin sediments at depths down to 345 mbsf[43].

Our results can be interpreted as evidence that evolution in the deep biosphere proceeds at extremely slow rates. Cells deep below the sediment surface must use available energy to maintain their cellular integrity over possibly geological timeframes while greatly attenuating cell division and genome replication[1,41], unless some physical disturbance or fluid flow returns them to the sediment surface. Under subsurface conditions, attenuated gene loss slows down the impact of selection that gradually shapes the subsurface biosphere[3]. However, the adaptive value for genes of motility and competition may be reduced with depth but is unlikely to expire entirely, since pore space constraints do not preclude slow microbial movement (millimeters over months), as demonstrated experimentally by gradual recolonization of deep subsurface sediments[44].

## Characteristics and distribution of dominant bacterial and archaeal groups
The Guaymas Basin subsurface yields predominantly MAGs affiliated with specific phylum-and order-level lineages that show distinct

mesophilic and thermophilic preferences. This suggests lineages appearing at specific depths and temperature ranges respond to environmental factors, which in turn shape their occurrence patterns. Our central working hypothesis is that the Guaymas Basin subsurface community is not a random assemblage, but reveals phylogenetic and functional structure that can be tracked downcore. Our account of this structured community focuses on dominant bacterial and archaeal phyla (Chloroflexota, Thermoproteota, Hadarchaeota); an extended overview on further bacterial and archaeal MAGs is provided in the Supplementary Note.

### Dominant subsurface bacteria

Of 63 bacterial MAGs found in the Guaymas Basin subsurface, 23 are members of the phylum Chloroflexota, one of the dominant phyla in marine sediments with metabolically diverse fermentative and dehalogenating lineages[45] (Supplementary Note). Within the Guaymas subsurface, Chloroflexota MAGs comprise 12 order-level lineages, and account for a significant fraction of recruited metagenomic reads per sample (up to 8.3%) (Fig. 2, Supplementary Fig. 6). At site U1545B, Chloroflexota MAGs were widespread within cool samples (2–20 °C) and persisted occasionally into deep and warm sediments; at Ringvent site U1547B they were ubiquitous in cool samples but also widely found in warm sediments (20–45 °C) (Fig. 2, Supplementary Fig. 6). MAGs that occur in warm sediments are affiliated with the subsurface and hydrothermal GIF9 group[31,46], the VGOG01 lineage from the sulfidic, warm water column of tropical Lake Tanganyika[47], and the dehalogenating Dehalococcoidales lineage. In hot sediments above 45 °C, Chloroflexota MAGs appear only in traces (Supplementary Fig. 6). Thus, the Guaymas Basin subsurface Chloroflexota generally prefer cool or moderately warm habitats, and avoid temperatures above ca. 40 °C.

Metagenomes were assembled and annotated for all Chloroflexota in our data sets to gather additional information about their metabolic potential (Supplementary Fig. 3, and Supplementary Dataset 8). Using the KEGG framework for functional annotation, within the general category "central carbohydrate metabolism" we find core genes that can participate in the TCA cycle, glycolysis, gluconeogenesis, and the pentose phosphate pathway (Supplementary Fig. 3). This category includes one specific module (K0378) that encodes an aldehyde ferrodoxin oxidoreductase (AOR), a tungsten-containing enzyme identified in mesophilic bacteria that can reduce aromatic compounds[48]. Within the category of "other carbohydrate metabolism" we find genes affiliated with galactonate/galactose degradation that can be linked to biosynthesis of alkaloids (e.g., terpenoid alkaloids). We also detect modules (assigned as "photorespiration") that are involved in the glycine cleavage system and shared between different amino acid biosynthetic pathways (Supplementary Note). In the general category "metabolic capacity" we detected genes assigned to oxygenic and anoxygenic photosynthesis, nonetheless, these are genes (e.g., pyruvate phosphate dikinase and citryl-CoA lyase) involved in carbon fixation. An expanded KEGG module analysis of the whole community metagenome (Supplementary Dataset 9) reveals many of the same genes, including those within the categories "central carbohydrate metabolism" and "other carbohydrate metabolism", reflecting the dominance of Chloroflexota among the recovered MAGs (Supplementary Fig. 4).

### Dominant subsurface archaea

Although the archaea contributed only 26 MAGs compared to 63 bacterial MAGs to our total, and represent fewer phylum-level lineages, they exhibit greater thermal range (Fig. 2). MAGS of two dominant archaeal phyla—the Thermoproteota (11 MAGs) and the Hadarchaeota (4 MAGs)—prefer warm and hot subsurface sediments, and are introduced here in greater detail; additional archaeal lineages are discussed in the Supplementary Note.

Archaeal MAGs were dominated by the Thermoproteota, an archaeal phylum consisting of four major lineages, the Thaumarchaeota, Aigarchaeota, Korarchaeota and Bathyarchaeia[49]. All 11 Thermoproteota MAGs belonged to the uncultured class Bathyarchaeia; these were detected at all examined sites but primarily at the Ringvent sites U1547B and U1548B (Fig. 2 and Supplementary Fig. 6). Order-level identification of bathyarchaeial MAGs reveals linkages to subsurface, seep and hydrothermal sediment habitats. Five MAGs assigned to the order-level lineages TCS64 and 40CM −2−53−6 were recovered primarily between 0.8-15 mbsf sediments at sites U1545B and U1547B with cool temperatures ranging from 2.8–17.4 °C. These bathyarchaeial orders have been reported also from deep sea brine pool samples[34] and from soil samples[50]. The order-level lineage B26-1, previously found in Guaymas Basin hydrothermal sediments[25,35], included three MAGs from warmer sediments (19–40 °C) below 63.8 mbsf at U1545B, and from warm to hot sediments (24–47 °C) between 19.3 and 65.8 mbsf at Ringvent site U1547B. The order-level lineage RBG-16-48-13, recovered previously from terrestrial subsurface cores[51], was represented by a MAG detected at site U1548 at 20–45 °C (Fig. 2). Two bathyarchaeial MAGs could not be classified at the order level, but one of these MAGs was abundant at temperatures between 39.5–47 °C at U1547B (Supplementary Fig. 6). The detection of bathyarchaeial MAGs over a wide temperature spectrum, and the link of bathyarchaeial orders to specific temperature regimes, suggests distinct thermal preferences among different lineages of Bathyarchaeia[52]. The ubiquitous presence of Bathyarchaeia in anaerobic sediments[53], including hydrothermal sediments[35], can be attributed to their capacity to metabolize multiple organic substrates, e.g., polysaccharides, urea, acetate, detrital proteins, and aromatics compounds such as benzoate and lignin[54], potential substrates in the hydrocarbon-rich Guaymas Basin subsurface. Based on MAG gene content, Bathyarchaiea can potentially utilize formaldehyde and shuttle it into carbon fixation via the Wood-Ljungdahl pathway (Supplementary Note). Lineage-specific thermophilic adaptations among the Bathyarchaeia include reverse DNA gyrase that facilitates DNA supercoiling under extreme temperatures[54].

Hadarchaeota thrive in subsurface sediments by a combination of heterotrophic traits (fermentation of carbohydrates) with autotrophic energy generation, specifically the oxidation of carbon monoxide and hydrogen[55]. Hadarchaeota were previously recovered from surficial hydrothermal sediments in Guaymas Basin[25]. Consistently, the 4 hadarchaeotal MAGs (GMP_008, GMP_020, GMP_027, GMP_034) did not recruit any reads from cool samples but only from warm and hot samples, indicating a preference for elevated temperatures (Fig. 2). In contrast to changing thermal preferences for MAGs from different bathyarchaeial orders, the Hadarchaeota, originally detected in hot and deep terrestrial aquifers[56], consistently prefer elevated temperatures in deep sediments of the Guaymas Basin subsurface (Fig. 2).

### Hadarchaeotal genomic features

Abundant and highly expressed hadarchaeotal MAGs were examined for characteristic features in their genomes and transcriptomes. One Hadarchaeota MAG, Guaymas_P_008, recruited ~19% of all metagenomic reads at 74.3 msbf (in-situ temperature 51 °C) at Ringvent site U1547B (Figs. 2, 3). This MAG contained genes for carbohydrate hydrolysis (α-RHA, β-galactosidase) and nucleoside uptake and degradation (nucleoside transporters, purine nucleosidases) that suggest purine/pyrimidine synthesis from nucleosides. This MAG also contained carbon monoxide oxidation genes (coxM, coxS) that were absent in the other Hadarchaeota MAGs that encoded genes for fermentation (porA, ack, acdA) and aromatics degradation (ubiX) (Supplementary Datasets 6, 7). The ability to utilize a wider range of carbohydrates may support higher temperature tolerance, as reported

for thermally-adapted Bathyarchaeia genomes[52]. The potential for hydrocarbon utilization in Hadarachaeota and other phyla (Supplementary Note) might contribute to reduced hydrocarbon concentrations at intermediate sediment depths and temperatures (Supplementary Fig. 1). One of our Hadarcheota MAGs (P_034) contained homologs to *mcrC* and *mcrG* that regulate the expression and assembly of the alkyl/methyl coenzyme M reductase operon[57], the essential methane and alkane-activating genes in archaeal methanogens, methane oxidizers and short-chain alkane oxidizers[58]. Finally, we note the presence of KaiC histidine in Hadarchaeota, a circadian clock protein that regulates cell division and allows prokaryotes to adapt to changes in environmental conditions[59], and the gene for programmed cell death (protein 5) that is linked to anti-virus defense and triggers dormancy under hostile conditions[60].

## Genome size trends in the deep biosphere

Comparisons of estimated genome sizes for all MAGs that recruited at least 0.1% of metagenomic reads from cool, warm, and hot sediments revealed a difference in average genome size. The most abundant genomes in cool sediments were on average significantly (two-sided partially overlapping samples *t*-test, adjusted $p < 0.05$) larger (-32%) than those recovered from hot sediments (Fig. 5A, B). The estimated genome size of MAGs recovered from our shallow (2–15 mbsf) samples was also ~22% larger on average than those detected in deeper (> 60 mbsf) and warmer (>30–40 °C) sediments. Linear regression analysis demonstrated a general reduction in average genome size in our samples as both temperature and depth increased (Fig. 5C, D). Elevated in situ temperatures are thought to select for smaller genome sizes via genome streamlining[61], for example increased gene loss after

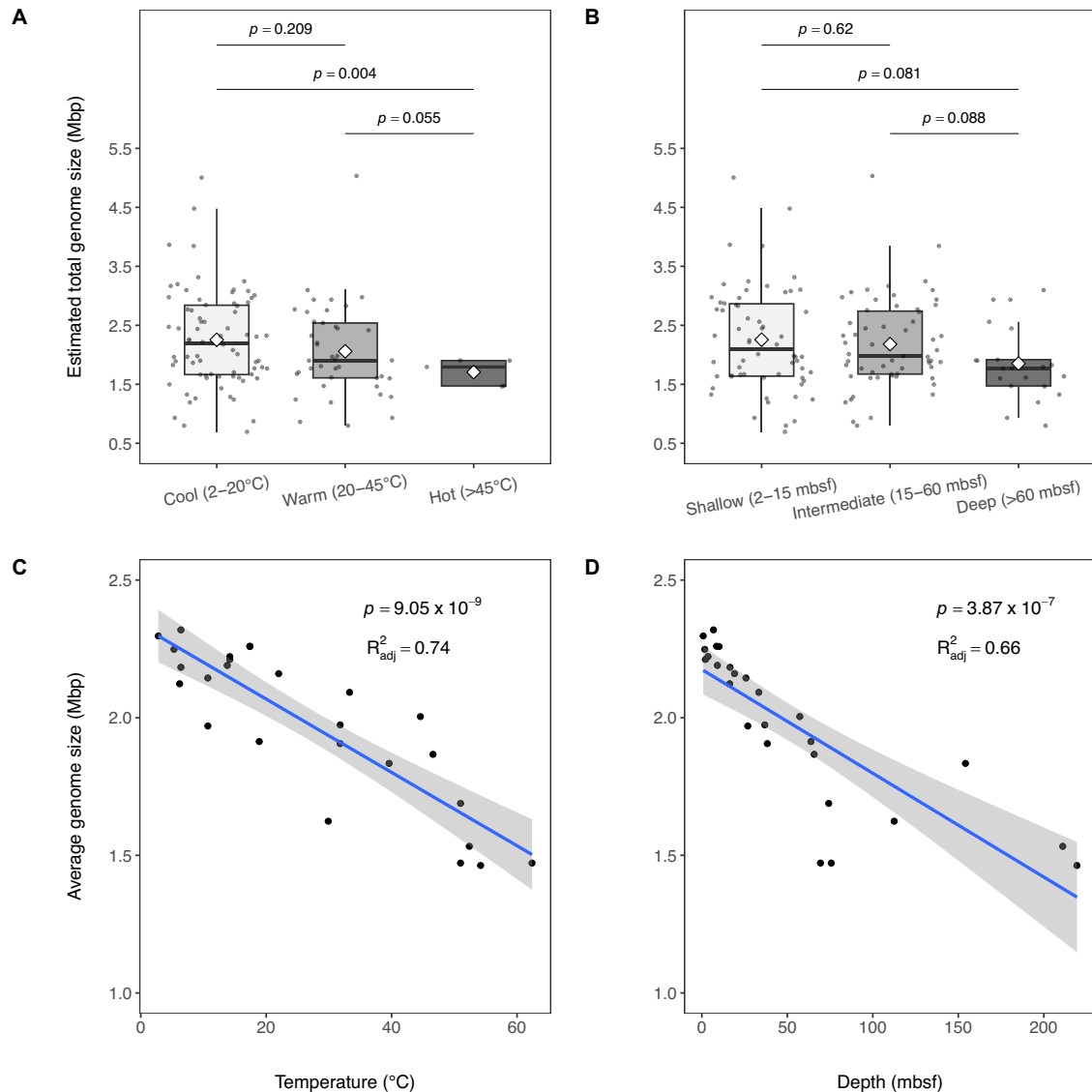

**Fig. 5 | Estimated and average genome size vs. temperature and depth.** Boxplots show estimated genome size of MAGs that recruited at least 0.1% of metagenomic reads from samples collected in cool (2–20 °C), warm (20–45 °C), and hot sediments (45–60 °C) in panel **A**, and at shallow (2–15 mbsf), intermediate (15-60 mbsf), and deep (>60 mbsf) depths in panel **B**. The Median is shown as the middle horizontal lines, the mean as the white diamonds, and interquartile ranges are shown as boxes (whiskers extend to 1.5 times the interquartile range). Each data point is overlaid on the boxplots and values at the top denote adjusted p-values from two-sided partially overlapping samples *t*-tests comparing estimated genome sizes by temperature (panel **A**) and depth (panel **B**) regimes. Panels **C** and **D** display the relationship between average estimated genome size in each metagenomic sample plotted against temperature (**C**) and depth (**D**) using linear regression. The blue lines in panels **C** and **D** denote the regression lines, with the fitted values +/− 1.96 standard error indicated by the grey bands. The values at the top of panels **C** and **D** denote the p-value and adjusted R-squared value of the fit. The Source Data file provides genome size estimates for MAGs contributing at least 0.1% of metagenomic reads for cool samples ($n = 74$), warm samples ($n = 41$), and hot samples ($n = 5$); genome size estimates for MAGs contributing at least 0.1% of metagenomic reads for shallow samples ($n = 62$), intermediate samples ($n = 59$) and deep samples ($n = 21$); and averaged MAG genome size estimates for each sample.

duplication; the effects of genomic streamlining are pervasive and result in the elimination of hundreds of genes all over the genome[62]. Reduced genome size lowers the metabolic cost required for microbial DNA replication, as suggested for hadal microorganisms in the Challenger Deep at Mariana Trench[63]. Microbes with smaller genomes would gain a relative survival advantage and gradually dominate the microbial community in the subsurface, as metabolically more demanding microbial community members with large genomes die off. Such a mechanism would contribute to the selection of subsurface-adapted microbial communities that has been documented already within the top few meters below seafloor[3], and it would explain the small size of microbial cells in deep subsurface sediments, near 0.5 micrometer[64].

### Temperature impact on MAG recovery

The environmental stresses that increasingly exclude microbial lineages, reduce genome size and reduce overall microbial population size (and thus, quantity of recovered DNA) are reflected in decreased recovery of MAGs in warmer and deeper samples from all sites (Fig. 6A, B). Plotted against depth, MAG recovery declines more quickly for the two hotter Ringvent sites U1547B and U1548B than for the cooler sites U1545B, U1546B and U1549B (Fig. 6B). When plotted against temperature, declining MAG recovery for the hot Ringvent sites and the cooler sites converge towards a shared minimum between ca. 50 and 60 °C (Fig. 6A). These comparisons show that the decline of MAG recovery with depth is locally modified, behaves differently at different sites, and does not follow a uniform depth-related decay rate. In contrast, the influence of increasing temperature is pervasive, reduces microbial diversity at all sites, and occludes the emergence of MAGs representing new microbial lineages beyond approximately 50–60 °C.

### Conclusions and outlook

While improved DNA and RNA recovery could potentially compensate for declining downcore cell density, and extend the recovery of new bacterial and archaeal MAGs towards deeper and hotter sediments, the observed trend towards increasingly limited microbial diversity in the subsurface stands in marked contrast to the numerous bacterial and archaeal lineages that thrive in surficial hydrothermal sediments of Guaymas Basin, where fluidized sediments are permeated by pulsating, extremely hot (> 80 °C) and highly reducing fluids[25]. We ascribe the difference to contrasting energy supply[65], and suggest that relatively moderate temperatures in IODP boreholes have a disproportionately greater impact on the energy-limited microbial deep biosphere, whereas surficial microbial communities that are well-supplied with energy-rich circulating hydrothermal fluids can tolerate high temperatures. The latter conditions select for thermophilic and hyperthermophilic, frequently chemolithoautotrophic bacteria and archaea[66,67]. We suggest that this difference ultimately results in distinct microbial communities in surficial hydrothermal sites, and in subsurface sediments where dominant bacteria and archaea (Chloroflexota, Thermoproteota, Acidobacteriota, Desulfobacterota) resemble the largely heterotrophic and mesophilic microbiota of non-hydrothermal benthic sediments[68,69].

Yet, we note that specific archaea, in particular the Hadarchaeota, show a preference for deep, hot sediments of Guaymas Basin. These archaea extend consistently into the deep sediment column, not only by MAG detection but also in 16S rRNA gene surveys[30], and appear to represent deep subsurface thermophiles that are sustained by substrates and energy sources of deep, hot sediments. Observations of microbial cells and activity in extremely deep, hot subsurface environments[5,70], could indicate such thermophile communities that have adapted to deep subsurface conditions. Since candidate archaea for deep, hot biosphere communities were consistently detected in the hydrothermally influenced Ringvent sites where a hot volcanic sill is emplaced into organic-rich marine sediments, we extrapolate further that the mineralogically and morphologically complex basalt interface[20] could provide microbial substrates and energy sources[71], calling for further studies.

## Methods
### Sample collection

Sediment cores were collected during IODP Expedition 385 using the drilling vessel *JOIDES Resolution*. Holes at each site were first advanced using advanced piston coring (APC), then half-length APC, and then extended core barrel (XCB) coring as necessary. Temperature

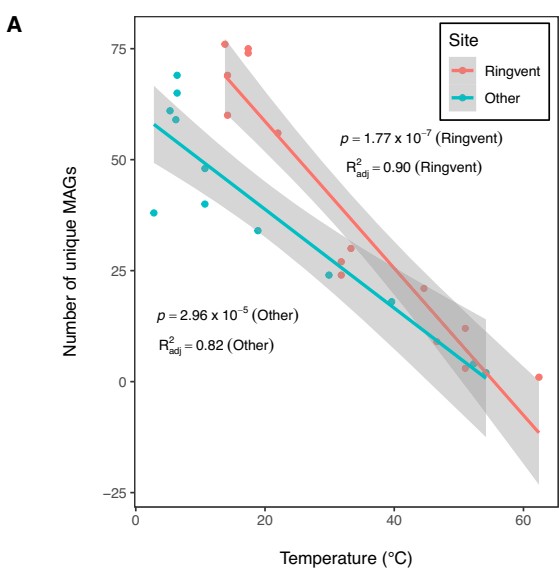

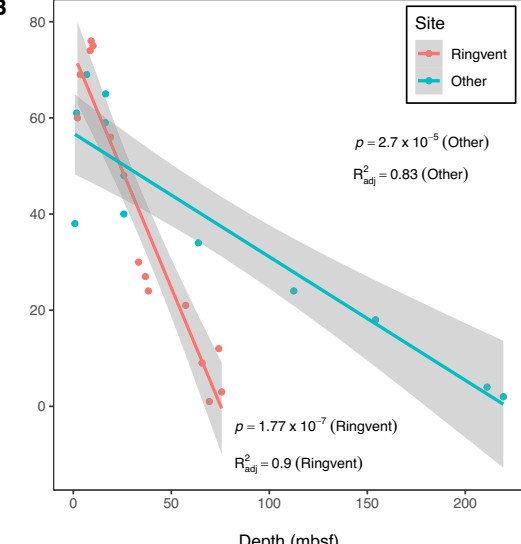

**Fig. 6 | MAG recovery at sampled sites vs. temperature and depth.** Panels **A** and **B** show the MAGs that recruited metagenomic reads from samples t sites U1547B and U1548B (in red) and sites U1545B, U1546B, and U1549B (in green) plotted against temperature (**A**) and depth (**B**) using best-fit linear regression. The solid lines in panels **A** and **B** denote the regression lines, with the fitted values +/− 1.96 standard error indicated by the grey bands. For both panels **A** and **B**, the *p*-values and adjusted R-squared values of the fits of each regression line are shown. MAG numbers for all samples ($n = 26$), and their depths and temperatures are provided in the Source Data file.

measurements used the advanced piston corer temperature (APCT-3) and Sediment Temperature 2 (SET2) tools[8]. Downhole logging conducted after coring used the triple combination and Formation MicroScanner sonic logging tool strings. After bringing core sections onto the core receiving platform of the D/V *JOIDES Resolution*, whole round samples for microbiology were retrieved within ~30 minutes using ethanol-cleaned spatulas. Samples for biogeochemical measurements were obtained and processed shipboard[17]. Whole round samples for DNA-based studies were capped with ethanol-sterilized endcaps, transferred to the microbiology laboratory, and stored briefly at 4 °C in heat-sealed tri-foil gas-tight laminated bags flushed with nitrogen until processing. Masks, gloves and laboratory coats were worn during sample handling in the laboratory where core samples were transferred from their gas-tight bags onto sterilized foil on the bench surface inside a Table KOACH T 500-F system, which creates an ISO Class I clean air environment (Koken Ltd., Japan). In addition, the bench surface was targeted with a fanless ionizer (Winstat BF2MA, Shishido Electrostatic Co., Ltd., Japan). Within this clean space, the exterior 2 cm of the extruded core section were removed using a sterilized ceramic knife. The core interior was transferred to sterile 50-mL Falcon tubes, labeled, and immediately frozen at −80 °C for post cruise analyses. For RNA-based studies, sampling occurred immediately after core retrieval on the core receiving platform by sub-coring with a sterile, cutoff 50cc syringe into the center of each freshly cut core section targeted. These sub-cores were immediately frozen in liquid nitrogen and stored at −80 °C.

### DNA extraction and sequencing
DNA was extracted from selected core samples using a FastDNA SPIN Kit for Soil (MP Biomedicals). Up to 5 grams of sediment were processed following a modified manufacturer's protocol[72]. Briefly, each sediment sample was homogenized twice (vs. once that the manufacturer suggests) in Lysing Matrix E tubes for 40 seconds at speed 5.5 m/s, using the MP biomedicals bench top homogenizer equipped with 2 ml tube adaptors. Between the two homogenization rounds the samples were placed on ice for 2 minutes. After the second homogenization the samples were centrifuged at 14,000 x *g* for 5 minutes. For each sample, the supernatant and the top layer of the pellet was transferred to a clean 2 ml tube where proteins were precipitated by the addition of the protein precipitation solution (PPS) provided in the extraction kit. The rest of the extraction protocol followed the manufacturer's recommendations. When parallel extractions were performed, the extracts were pooled and concentrated using EMD 3kDa Amicon Ultra-0.5 ml Centrifugal Filters (Millipore Sigma). A control extraction, in which no sediment was added, was included to account for any laboratory contaminants (Supplementary Materials). All libraries for metagenome sequencing (n = 29; 26 samples and 3 controls; Supplementary Data 2) were prepared from genomic DNA extracts that were submitted at the University of Delaware DNA Sequencing & Genotyping Center. Thirteen libraries were sequenced with NovaSeq S4 PE150 (Illumina) at the University of California, Davis Genome Center, and thirteen libraries were sequenced with NextSeq550 (Illumina) at the University of Delaware DNA Sequencing & Genotyping Center. Metagenome sequence reads were deposited to the National Center for Biotechnology Information Sequence Read Archive under access numbers SRR23614663-23614677 and SRR22580794-SRR22580807 (Bioproject PRJNA909197).

### Metagenomic co-assembly, binning, dereplication and taxonomic assignment
Metagenomic reads originating from adjacent regions (such as adjacent depths targeted in this study) are likely to overlap in the sequence space, increasing the mean coverage and extent of reconstruction of MAGs when using a co-assembly approach. Before assembly, reads were trimmed for quality and adapters removed using Trimmomatic

v0.39[73] (parameters: leading:20; trailing:20; sliding window: 0-24; min length 50). The quality of reads was verified with FastQC v0.11.9[74]. For MAG reconstructions, we used the trimmed reads of metagenomic datasets from all 29 Guaymas samples sequenced in this study (Supplementary Data 2). The 26 metagenomes were co-assembled into contigs using MEGAHIT 1.2.9[75] with default parameters. For determining non-redundant MAGs, assembled contigs were binned using three different binners, MetaBAT2 2.12.183[76], MaxBin2 2.2.7[77], as well as CONCOCT 1.1.0[78]. Output bins from all three binning algorithms were refined and dereplicated using DAS Tool 1.1.6[79]. DasTool determines a unique MAG through a single-copy gene (SCG) scoring strategy coupled to an iterative bin de-replication procedure that produces the highest-scoring set of non-redundant bins (in terms of SGC completeness/contamination) from input bins generated by different binners. Completeness, size, and contamination levels of the reconstructed genomes were estimated using CheckM2 1.0.0[26]. Only MAGs that were at least 50% complete and contained less than 10% contamination were used for downstream analyses (Supplementary Data 4). The taxonomic placement of the MAGs was performed with GTDB-Tk 2.1.0[80].

To account for seawater and laboratory contamination (Supplementary Note), control samples (Supplementary Data 2) identified MAGs of lab/control contaminants, including Patescibacteria (Paceibacteria, Microgenomatia), Actinobacteriota (Actinomycetia, Humimicrobia), Gammaproteobacteria (Pseudomonadales, Burkholderiales), and Firmicutes (Staphylococcales); these were removed from downstream analyses (Supplementary Data 3).

### Calculation of MAG relative abundances
Metagenomic reads from 26 samples were mapped to each MAG using the CoverM 0.6.1 (https://github.com/wwood/CoverM) command line tool with the BWA 2.0 aligner[81]. The CoverM tool automatically concatenated all the MAGs into a single file, and metagenomic reads were recruited to MAG contigs, setting the parameter --min-read-percent-identity to 95 and --min-read-aligned-percent to 50. The "Relative Abundance" CoverM method on the "genome" setting was used to calculate the percent of total metagenomic reads per sample that mapped to each of the 89 MAGs. A custom R script was utilized to concatenate all coverM output files into a single file in a matrix format (with each sample representing a column and each row representing total percent of DNA-Seq reads per sample that mapped to a MAG) and was used for heatmap plotting.

### Gene annotation, and prediction of KEGG metabolic module presence/absence using MetaPathPredict
Genes were called for all MAGs using Prodigal 2.6.3[82] and then annotated using Prokka 1.14.6[83], KofamScan 1.3.0[84], and METABOLIC 4.0[85] using default settings. KEGG modules for bacterial MAGs were reconstructed using gene annotations from the KofamScan 1.3.0 command line tool, and the presence or absence of incomplete modules in the genomes was predicted using MetaPathPredict 1.0.0[86] with default settings. MetaPathPredict cannot yet be applied to archaeal MAGs. Briefly, Prodigal was used to call genes, and KofamScan was used to annotate them. Gene annotations were generated for predicted genes from bacterial MAGs, and were used as input to MetaPathPredict, which generated predictions for the presence or absence of KEGG modules based on the gene annotations of all bacterial MAGs.

### CCA and nMDS analyses of metagenomic abundance datasets and associated environmental parameters
The abundances of metagenomic reads mapped to MAGs were normalized using the "transcripts per million" normalization method[87] with the read mapping "counts" output from coverM (https://github.com/wwood/CoverM). The abundance data were analyzed using canonical correlation analysis (CCA) as well as non-metric

multidimensional scaling (nMDS) and were fitted with the environmental parameters in Supplementary Data 1 using R[88]. The cca and metaMDS functions were used for CCA and nMDS analyses, respectively, as well as the envfit function from the vegan 2.6-4 package[89]. The results were plotted using ggplot2 3.3.6[90] with sample shapes corresponding to temperature regime.

## Estimated genome size analysis

The estimated genome size of all 89 MAGs was calculated by dividing the MAG assembly size (total base pair length of the MAG) by the fractional CheckM2 completeness of the MAG (the default CheckM2 completeness output divided by 100; a number between 0 and 1). Difference in genome size distributions for MAGs that recruited at least 0.1% of metagenomic reads from samples across temperature (cool [2–20 °C], warm [20–45 °C], hot [45–62 °C]) and depth (shallow [2–15 mbsf], intermediate [15–60 mbsf], deep [>60 mbsf]) regimes was assessed using the two-sided partially overlapping samples *t*-test[91], and resulting *p*-values were adjusted for multiple comparisons via Benjamini-Hochberg correction. The average estimated genome size of MAGs that recruited at least 0.1% of reads from metagenomic samples ($n = 26$) was fitted using linear regression against temperature and depth measurements affiliated with the samples.

## MAG recovery at sampled sites versus temperature and depth

The number of non-redundant MAGs that recruited at least 0.1% of reads from metagenomic samples ($n = 26$) was fitted using linear regression against temperature and depth measurements affiliated with the samples. Temperature values were interpolated for each sample using linear regression of the local thermal gradient (°C/m) multiplied by depth (mbsf), plus the y-axis intercept: U1545B, T = 0.225 x depth + 4.899; U1546B, T = 0.221 x depth + 2.627; U1547B, T = 0.511 x depth + 13.01; U1548B, T = 0.804 x depth + 6.499; U1549A/B, T = 0.194 x depth + 3.532.

## Scanning of MAGs for secondary metabolite biosynthetic gene clusters

All 89 MAGs were individually scanned for secondary metabolic biosynthetic gene clusters using antiSMASH 6.0[92] with default parameters. Resulting gene cluster prediction results (in GenBank format) were parsed and their gene content was analyzed. Clusters with a total length less than 5kb were discarded from downstream analysis to minimize the inclusion of fragmented biosynthetic clusters in the analysis.

## RNA extraction, library preparation, sequencing, and mapping of RNA reads to the MAGs

RNA was extracted from 19 sediment samples from sites U1545B-U1552B and a blank sample (control) using the RNeasy PowerSoil Total RNA Kit (Qiagen) following the manufacturer's protocol with modifications which are discussed below. RNA samples were prepared from samples spanning the depths 0.8 to 101.9 mbsf. All samples, including a blank control, were first washed twice with absolute ethanol (200 proof; purity ≥ 99.5%), and sterile DEPC water (once) to reduce hydrocarbons and other inhibitory elements that otherwise resulted in low RNA yield. In brief, 13-15 grams of frozen sediments were transferred into UV-sterilized 50 ml Falcon tubes (RNAase/DNase free) using clean, autoclaved and ethanol-washed metallic spatulas. Each sample transferred into the 50 ml Falcon tube received an equal volume of absolute ethanol and was shaken manually for 2 min followed by 30 seconds of vortexing at full speed to create a slurry. Samples were spun in an Eppendorf centrifuge (5810R) for 2 minutes at 2000 x *g*. The supernatant was decanted and after the second wash with absolute ethanol, an equal volume of DEPC water was added into each sample and samples were spun for 2 minutes at 2000 x *g*. The supernatant was decanted, and each sediment sample was immediately divided into three 15 mL Falcon tubes containing beads provided in the PowerSoil Total RNA Isolation Kit (Qiagen). The RNA extraction protocol was followed as suggested by the manufacturer, with the modification that the RNA extracted from the three aliquots was pooled into one RNA collection column. All steps were performed in a UV-sterilized clean hood equipped with HEPA filters. Surfaces inside the hood and pipettes were thoroughly cleaned with RNase AWAY™ (Thermo Scientific™) before every RNA extraction and in between extraction steps.

Trace DNA contaminants were removed from RNA extracts using TURBO DNase (Thermo Fisher Scientific) and the manufacturer's protocol. Removal of DNA was confirmed by negative PCR reactions using the bacterial primers BACT1369F/PROK1541R (F: 5′CGGTGAATACGTTCYCGG 3′, R: 5′AAGGAGGTGATCCRGCCGCA 3′) targeting the 16S rRNA gene[93]. Each 25 µl PCR reaction was prepared using 0.5 U µl$^{-1}$ GoTaq® G2 Flexi DNA Polymerase (Promega), 1X Colorless GoTaq® Flexi Buffer, 2.5 mM MgCl$_2$, (Promega) 0.4 mM dNTP Mix (Promega), 4 µM of each primer (final concentrations), and DEPC water. PCR reactions used an Eppendorf Mastercycler Pro S Vapoprotect (Model 6321) thermocycler with following conditions: 94 °C for 5 min, followed by 35 cycles of 94 °C (30 s), 55 °C (30 s), and 72 °C (45 s). The PCR products were run in 2% agarose gels (Low-EEO/Multi-Purpose/Molecular Biology Grade Fisher BioReagents™) to confirm absence of DNA amplification. Amplified cDNAs from the DNA-free RNA extracts were prepared using the Ovation RNA-Seq System V2 (Tecan) following manufacturer's suggestions. All steps through cDNA preparation were completed the same day to avoid freeze/thaw cycles. cDNAs were submitted to the Georgia Genomics and Bioinformatics Core for sequencing using NextSeq 500 PE 150 High Output (Illumina). The cDNA library generated from our control did not contain detectable DNA. It was nonetheless submitted for sequencing, but it failed to generate any sequences that met the minimum length criterion of 300-400 base pairs.

Reads from the 13 metatranscriptome samples collected from sites that metagenomic samples were taken from were mapped to each MAG using the CoverM 0.6.1 (https://github.com/wwood/CoverM) command line tool with the BWA 2.0 aligner[81]. The CoverM tool automatically concatenated all the MAGs into a single file, and metatranscriptome reads were recruited to MAG contigs, setting the parameter --min-read-percent-identity to 95 and --min-read-aligned-percent to 50. A custom R script was utilized to concatenate all coverM output files into a single file in a matrix format, with each sample representing a column and each row representing total percent of RNA-Seq reads per sample that mapped to a MAG. The output was used in this study for heatmap plotting to examine evidence for activity of the taxa for which we recovered MAGs. Metatranscriptome reads were deposited to the National Center for Biotechnology Information Sequence Read Archive under accession numbers SRR22580929-SRR22580947 (Bioproject PRJNA909197).

## Cell counts

The sediment sampling for cell counts occurred immediately after core retrieval on the core receiving platform by sub-coring with a sterile, tip-cut 2.5 cc syringe from the center of each freshly cut core section. Approximately 2 cm$^3$ sub-cores were immediately put into tubes containing fixation solution consisting of 8 mL of 3xPBS (Gibco™ PBS, pH 7.4, Fischer) and 5% (v/v) neutralized formalin (Thermo Scientific™ Shandon™ Formal-Fixx™ Neutral Buffered Formalin). If necessary, the mixture was stored at 4 °C.

Fixed cells were separated from the slurry using ultrasonication and density gradient centrifugation[94]. For cell detachment, a 1 mL aliquot of the formalin-fixed sediment slurry was amended with 1.4 mL of 2.5% NaCl, 300 µL of pure methanol, and 300 µL of detergent mix[95], 100 mM ethylenediamine tetraacetic acid [EDTA], 100 mM sodium pyrophosphate, 1% [v/v] Tween-80). The mixture was thoroughly shaken for 60 min (Shake Master, Bio Medical Science, Japan), and

subsequently sonicated at 160 W for 30 s for 10 cycles (Bioruptor UCD-250HSA; Cosmo Bio, Japan). The detached cells were recovered by centrifugation based on the density difference of microbial cells and sediment particles, which allows collection of microbial cells in a low-density layer. The sample was transferred onto a set of four density layers composed of 30% Nycodenz (1.15 g cm$^{-3}$), 50% Nycodenz (1.25 g cm$^{-3}$), 80% Nycodenz (1.42 g cm$^{-3}$), and 67% sodium polytungstate (2.08 g cm$^{-3}$). Cells and sediment particles were separated by centrifugation at 10,000 × $g$ for 1 h at 25 °C. The light density layer was collected using a 20G needle syringe. The heavy fraction, including precipitated sediment particles, was resuspended with 5 mL of 2.5% NaCl, and centrifuged at 5000 × $g$ for 15 min at 25 °C. The supernatant was combined with the previously recovered light density fraction. With the remaining sediment pellet, the density separation was repeated. The sediment was resuspended using 2.1 mL of 2.5% NaCl, 300 μL of methanol, and 300 μL of detergent mix and shaken at 500 rpm for 60 min at 25 °C, before the slurry sample was transferred into a fresh centrifugation tube where it was layered onto another density gradient and separated by centrifugation just as before. The light density layer was collected using a 20G needle syringe, and combined with the previously collected light density fraction and supernatant to form a single suspension for cell counting.

For cell enumeration, a 50%-aliquot of the collected cell suspension was passed through a 0.22-μm polycarbonate membrane filter. Cells on the membrane filter were treated with SYBR Green I nucleic acid staining solution (1/40 of the stock concentration of SYBR Green I diluted in Tris-EDTA [TE] buffer). The number of SYBR Green I– stained cells were enumerated either by direct microscopic counts[70] or image-based discriminative counts[96]. For image-based discriminative counting, the Count Nuclei function of the MetaMorph software (Molecular Devices) was used to detect and enumerate microbial cells.

### Reporting summary

Further information on research design is available in the Nature Portfolio Reporting Summary linked to this article.

## Data availability

The raw metagenome and metatranscriptome sequence data generated in this study have been deposited in the NCBI GenBank database under the Bioproject accession number PRJNA909197. Metatranscriptome reads are deposited under the accession numbers SRR22580929-SRR22580947. Metagenome reads are deposited under the accession numbers SRR22580794-SRR2258807 and SRR23614663-SRR236114677. Biogeochemical and thermal shipboard data for all IODP385 sites discussed in this study (U1545-U1552) are publicly available on the IODP Expedition 385 online report (http://publications.iodp.org/proceedings/385/385title.html). Shipboard data can be downloaded for each drilling site individually, as numbered excel tables. Post-cruise geochemical data sets (DIC, TOC, TN, hydrocarbons) have been submitted to the Biological and Chemical Oceanography database (BCO-DMO) and are publicly available under project number 833856 (https://www.bco-dmo.org/project/833856). Publicly available datasets used in this study include the CheckM2 database (https://zenodo.org/record/4626519), the GTDB-Tk database release R214 (https://ecogenomics.github.io/GTDBTk/installing/index.html), the KOfam database (ftp://ftp.genome.jp/pub/db/kofam/), the METABOLIC database (https://github.com/AnantharamanLab/METABOLIC), the MEROPS database (https://www.ebi.ac.uk/merops/download_list.shtml), the dbCAN2 database (http://bcb.unl.edu/dbCAN2/download/Databases/dbCAN-old@UGA/dbCAN-fam-HMMs.txt), ISfinder database (https://isfinder.biotoul.fr/), NCBI Bacterial Antimicrobial Resistance database (https://www.ncbi.nlm.nih.gov/bioproject/313047), UniProtKB (SwissProt) database (https://www.uniprot.org/uniprot/?query=reviewed:yes), Prokka databases (https://github.com/tseemann/prokka) and the antiSMASH 6.0

databases (https://dl.secondarymetabolites.org/releases/). Source data are provided with this paper.

## Code availability

All custom scripts used for data analysis and figure creation are available in the GitHub repository at https://github.com/d-mcgrath/guaymas_basin[97].

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

## Acknowledgements

The authors would like to acknowledge the crew and entire science party for IODP Expedition 385 for their assistance with sample collection. Without their assistance this study would be impossible. The authors would also like to thank Gustavo Ramírez for his assistance with DNA extractions using his method. We thank Mark Shaw and Bruce Kingham in the University of Delaware DNA Sequencing & Genotyping Center for assistance with sample preparation and Illumina sequencing. This study was supported by NSF Grant OCE-2046799 to VE, PM, AT, and R. Hat-zenpichler, by NSF grant OCE-1829903 to VE, PM, and AT, and by JSPS KAKENHI Grants JP19H00730, JP22K18426, and JP23H00154 to YM.

## Author contributions

V.E., A.T., and P.M. designed the study. V.E. and YM took primary responsibility for collecting the samples during IODP Expedition 385 together with shipboard scientists. A.T. served as Co-Chief Scientist of IODP Expedition 385. D.B. and P.M. extracted DNA and RNA for metagenomes and metatranscriptomes, respectively. D.B. prepared

metagenome libraries for sequencing, and handled data deposition into GenBank. DGM took primary responsibility for bioinformatic processing of metagenome data and mapping of transcripts to MAGs. P.M. and D.G.M. analyzed the MAG data and VE and AT contributed to data interpretation. Y.M. provided cell count data. P.M., D.G.M., V.E., and A.T. co-wrote the first draft of the manuscript. A.T. and P.M. led writing of all subsequent drafts, and all authors contributed to its final form.

## Competing interests

The authors declare no competing interests.
