## [Peer Review File · Nature Communications]

Metagenomic Profiles of Archaea and Bacteria within Thermal and Geochemical Gradients of the Guaymas Basin Deep SubsurfaceReviewer #1 (Remarks to the Author):

Geller-McGrath et al. used metagenomic and metatranscriptomic approaches to study the environmental distribution, function and activity patterns of bacteria and archaea along thermal, geochemical, and cell count gradients in five different drilling sites of the Guaymas Basin (Gulf of California, Mexico). The authors found that the composition and distribution of MAGs dominated by Chloroflexota and Thermoproteota were determined by biogeochemical parameters in sediment layers with moderate temperature (lower than 45°C), but become influenced by temperature as it exceeds 45°C. Overall, MAG genome sizes and diversity decreased with increasing temperature. Additionally, the study revealed that specific microbial populations of Thermoproteota and Hadarchaeota showed an increasing trend in relative abundance and recruitment of reads from the transcriptome with increasing depth temperature.

The strength of the manuscript is the systematic analysis of the biogeochemical parameters along with the genomes and metatranscriptomes of the microorganisms in the deep subsurface of hydrothermal regions. However, there are several key shortcomings that greatly limited the quality of the MS, including: (1) lacking of enough details on methodologies, especially detailed information about the omics, which lowered the possibilities to evaluate the quality of the manuscript. For example, The concentration, total amount and integrity of the DNA and RNA for omics sequencing? How did the metagenome and metatranscriptome libraries been constructed? What are the sequencing depths? (2) The scientific question of the manuscript is not quite clear (or not specific enough). The major body of the manuscript is mainly descriptive information, and the results and discussion focused too much on scattered details and lack integration and in-depth analysis. (3) Additionally, the significance and impact of the findings were not clearly discussed and illustrated, which need to be further elaborated.

Some detailed comments:

Introduction: The introduction on biodiversity, activity and functions of microbial communities of the Guaymas Basin, are not sufficient to allow readers to understand the significance and necessity of this study; and the scientific question and objectives of the study were not clearly stated.

Line 78-79, this sentence is not clear. what exactly do you mean by "extent and characteristics" ?

Line 92-93, this seems to be the scientific question the MS wanted to address, however, there were too many questions within this sentence, e.g., environmental distribution, genomic potential, adaptation, stress tolerance. Although each of the above questions was mentioned in the manuscript, none of them were clearly and solidly illustrated. I would suggest the authors to be more focus on one or two points and make in-depth analysis and discussion.

Line 129, what do you mean by "incomplete downcore sulfate depletion"?

Line 142-143, the authors measured petroleum hydrocarbon, why not involving them in later biostatistical analysis?

Line 148-149, why did negative controls have sequenced reads and matched with MAGs?

Line 194-213, As one of the major conclusion in the manuscript, the influence of environmental factors on MAG composition is largely descriptive and lacking of sufficient biostatistical supporting. For example, Line 203-206, the authors claimed different environmental parameters that affected the MAG composition in different temperature zone. However, the results were mainly based on the descriptions on nMDS plot, which was lacking of direct statistical support. Also, the authors only described the parameters with positive correlation with the MAG composition and ignored those with negative correlations. In addition, the validity of using nMDS analysis to study the influence of environmental parameters need to be evaluated. Why not using CCA, RDA etc. ?

Line 215-246, the "Metagenomic subsurface adaptation". (1) What are environmental features of

studied sites that are special and distinct from other marine habitats? How does the annotated functions provide adaptation advantage for the MAGs, associated with the special environmental conditions of the studied sites? The linkage between the genomic features and specific environmental condition of the study sites is weak. (2) The identified "adaptation features" seemed to be commonly present in other marine habitats such as marine sediments, so what are the special adaptation mechanisms for the thermally heated subsurface microorganisms?

Line 239-240, how common did the related functions in the MAGs (percentage)?

Line 273-276, this sentence reads quite strange, please modify.

Line 358 and other locations, what do you mean by "unique MAGs"? A definition is needed.

Line 357-367, the relationship between MAGs recovery and temperature may need to be revisited by considering the sequencing depth of the metagenomes. As the variation on sequencing depth would greatly impact the MAG recovery rate and reads recruitment. For example, if the metagenomes from the hot regions were generally with less sequencing depth (smaller data size), the observed trends of decreasing MAG recovery with temperature would be questionable.

Line 424-425, More details in library construction are needed. Were they whole genome amplified before sequencing? What were the concentration and total amount of DNA extracted from the samples? What is the sequencing depth? How many Gb of data for each metagenome?

Line 445-449, Why were there contaminants ? and how could they been sequenced and with so many reads?

Line 475, not clear. How could abundance of metagenomic reads been normalized as "transcripts" per million?

Line 536-537, Please provide the concentration and integrity of the RNAs extracted. Also, provide details on cDNA libraries construction. Were the RNA samples with enough concentration and total amount for transcriptome sequencing?

Figure 4, Description is not clear. Are those data based on composition of MAGs or entire metagenomes from each site? Also, the resolution of the plot was low.

The methodologies for biogeochemical parameter measurement seemed to be missing?

Reviewer #2 (Remarks to the Author):

In the manuscript "Metagenomic Profiles of Archaea and Bacteria within Thermal and Geochemical Gradients of the Guaymas Basin Deep Subsurface" Geller-McGrath and Mara et al present a genomic resolution of microbes and their transcripts along a temperature gradient in the Guaymas Basin. The authors combine geochemical measurements with genome-resolved metagenomics and transcriptomics. While I think this paper is of merit and it definitely has the potential to be appear in Nature Communications, I do have some reservations regarding the presentation of the results:

1. The manuscript is very technical. While the results are outstanding and interesting per se, I'm missing an interpretation that makes these findings important, e.g. ecosystem functioning. Even the title is very descriptive. So, the story, the meaning of the results need to be developed.
2. The headers of the results/discussion section should present a major finding rather than be

technical and – in the worst case – state a method or simply a phylum name like “Chloroflexota”, which doesn’t tell the reader anything.

3. Although I do like the correlation of MAG recovery/size to temperature or depth, I’d love to see how the functions in the ecosystem are maintained in the metagenomes (not only binned genomes) and metatranscriptomes (mapped to entire (assembled) metagenomes).

4. How do the results of the current study compare to “Rapid metabolism fosters microbial survival in the deep, hot subseafloor biosphere” by Beulig et al. Nature Communications volume 13, Article number: 312 (2022)? I’m missing this comparison because it seems that the current study contradicts the findings of Beulig et al.

Minor comments:

L 36 and elsewhere in the manuscript: It should be “By contrast, xxx” or “In contrast to XY, xxx” but not “In contrast, xxx”.

L 95: consider rephrasing “mRNA evidence”.

L199-203: Have the authors considered using a bioENV to figure out the best combination of environmental variables that described the community composition?

L336: Have the authors considered genome completeness of the MAGs? Taking the completeness into account when it comes genome sizes might be important but a metric other than CheckM needs to be used, for which markers are equally distributed across the genome or generally takes the unequal distribution into account.

L369: How can DNA/RNA recovery overcome cell density? I do not understand the ecological context here.

L431: Have the authors considered combined assemblies of metagenomes and metatranscriptomes to promote recruitment of reads? I’m not saying this is a necessity, but it could improve the mapping frequency. What is the rationale behind co-assembling all the 26 metagenomes rather than individually assembling the metagenomes?

Figure 1: In-situ should be italics (in figure and in legend)

Figure 6: MAG recovery can be affected by many biological features (e.g., strain heterogeneity). I think a different metric for diversity should be used, which is based on marker genes like ribosomal protein S3 to identify if temperature and depth have an impact on the recovery of genomes / diversity per se.

Reviewer #3 (Remarks to the Author):

In “Metagenomic profiles of archaea and bacteria within thermal and geochemical gradients of the Guaymas Basin deep subsurface,” the authors examine trends in microbial community composition and metabolic potential by examining several MAGs recovered from cores at various points in Guaymas Basin.

Overall, the manuscript is well-written and the methods are sound. My primary piece of feedback is more of a question: why focus only on MAGs for this study rather than also include results from the full metagenomes? As the authors note, there are limitations and perhaps biases to MAG recovery that result from challenges like poor assembly, poor read recruitment, etc, and it is possible that these limitations may introduce biases with depth, skewing the results. It seems to me that including some results from an analysis of the full metagenome (i.e. a broader survey of community composition using something like mmseqs2, and/or a broader survey of total metabolic potential focusing on relative abundance of key metabolic genes independent of MAGs) would provide broader context for how the whole community changes with depth independent of the MAGs, and would ameliorate potential biases in MAG recovery that may occur with depth.

In addition to this, the authors put a lot of information in the supplement, but some parts of the Results/Discussion feel overly broad and/or incomplete. I would suggest adding some details back into the Results/Discussion, especially as it pertains to specific metabolic potential of specific taxa/bins and how that changes with depth. As a further example, Supplementary Figure 2

appears to simply be a copy of Figure 2, with a bit more taxonomic detail. It seems to me that these details should be included in the main Figure 2 and there doesn't need to be an additional figure in the supplement.

Further comments are listed below.

Introduction

Line 48: maybe clarify that this would be downcore in cool/moderate temps, right? These are subsurface sediments away from hydrothermal activity?

Methods:

Line 435 What read trimming technique was used?

This is just a suggestion, but if the authors are trying to reconstruct metabolic modules and the pipelines they use can't do archaea, perhaps try KEGGdecoder, which is more agnostic.

For the most part, though, the methods make sense and it seems that appropriate controls were run and appropriate precautions were taken regarding contamination.

Results and Discussion:

Line 116: remove "at": "cell counts decreased by four to five orders of magnitude"

For the section on Subsurface biogeochemical zonation: a lot of chemistry data is reported in table 1. This is a bit hard to visualize in context of samples—would it be possible to show this as a figure somehow?

It is unclear to me what units are in the x-axis of supplementary figure 1.

For the section on MAG diversity, distribution and evidence of activity:

The authors comment on relative abundances of MAGs in different depths, which is interesting, but why are specific MAGs within a specific phylum more abundant than others? More description of differences between specific MAGs (sort of like a mini pangenome analysis) would help contextualize these results. For example, why are some Chloroflexi at high abundance and not others? Is there information that might contextualize this based on the metabolic pathway completeness analyses?

Along the lines of the above comment, a figure showing bin metabolic pathway completeness heatmap to compare with figures 2 and 3 would be helpful. (KEGGDecoder makes heatmaps like these, but I am sure other pipelines do too.)

Lines 171-178: The section on metabolisms, and the mention of CRISPRs, needs to be contextualized, as it is seemingly thrown in there with no further discussion. What can we derive from this?

Figure 2: For relative abundance, did the authors normalize according to the length of the MAG? The values are reported as what percent of reads per sample mapped to a MAG, so it's normalized for the number of reads per sample, but in addition to that it seems like MAGs that are longer would have more reads that recruit to them. I checked the Materials and Methods and this isn't clear—is this normalization calculated as part of CoverM?

For the section on metatranscriptome mapping (lines 179-194):

As above, this section would benefit from a bit more detail and context. Why are specific MAGs more active than others within specific phyla? Were certain metabolisms more active than others as depth and/or temperature increased? Were these metabolisms specific to certain phyla? How did activity compare with overall DNA abundance?

In examining the supplementary material, it seems that a lot of this information is in the supplementary material, but without this context in the main manuscript, the results are overly vague. Please consider perhaps condensing much of the info in the supplementary materials and putting it in the main manuscript.

Add "that" to line 219: "widespread genomic features that extend"

Lines 241-248: this is an interesting result; the authors should have the data indicating whether genes for chemotaxis were actually expressed in the MAGs, no? It seems that would be worth reporting here.

Supplementary Figure 2 seems like it's exactly figure 2, but with more detail regarding which taxa each row correspond to. It seems like that should be included in figure 2, and supplementary figure 2 could be dispensed with.

Consider reorganizing the text: most of the phylum-specific MAG description is focused on abundance (DNA), not RNA. One suggestion would be to add the metatranscriptomic results later, and/or add metatranscriptomic results to phylum-specific descriptions.

As I mentioned above, I also think this section would be helped by a metabolic pathway completion map like those produced by KEGGdecoder. Would help visualize differences between these MAGs, visualize metabolic potential.

Rika Anderson

REVIEWER COMMENTS

Reviewer #1 (Remarks to the Author):

Geller-McGrath et al. used metagenomic and metatranscriptomic approaches to study the environmental distribution, function and activity patterns of bacteria and archaea along thermal, geochemical, and cell count gradients in five different drilling sites of the Guaymas Basin (Gulf of California, Mexico). The authors found that the composition and distribution of MAGs dominated by Chloroflexota and Thermoproteota were determined by biogeochemical parameters in sediment layers with moderate temperature (lower than 45°C), but become influenced by temperature as it exceeds 45°C. Overall, MAG genome sizes and diversity decreased with increasing temperature. Additionally, the study revealed that specific microbial populations of Thermoproteota and Hadarchaeota showed an increasing trend in relative abundance and recruitment of reads from the transcriptome with increasing depth temperature.

Response: This are indeed the major points of our manuscript, well summarized.

The strength of the manuscript is the systematic analysis of the biogeochemical parameters along with the genomes and metranscriptomes of the microorganisms in the deep subsurface of hydrothermal regions. However, there are several key shortcomings greatly limited the quality of the MS, including: (1) lacking of enough details on methodologies, especially detailed information about the omics, which lowered the possibilities to evaluate the quality of the manuscript. For example, The concentration, total amount and integrity of the DNA and RNA for omics sequencing? How did the metagenome and metatranscriptome libraries been constructed? What are the sequencing depths? (2) The scientific question of the manuscript is not quite clear (or not specific enough). The major body of the manuscript is mainly descriptive information, and the results and discussion focused too much on scattered details and lack integration and indepth analysis. (3) Additionally, the significance and impact of the findings were not clearly discussed and illustrated, which need to be further elaborated.

Response: Thank you for identifying three major areas of improvement. In response to 1), we are now reporting DNA and RNA yield, and corresponding libraries (before and after QC) in a new table (Supplementary Table 2). We are reporting methodologies in greater depth (lines 474-475; 502-523; 621-625). 2) The scientific rationale for metagenomic exploration of the deep biosphere was explained more explicitly in the revised introduction (new lines 51 to 56), and the manuscript text is revised extensively to reveal the overall scientific perspective underlying the manifold findings that we are reporting (as recommended by other reviewers as well). 3) We have developed the Conclusion section further (now "Conclusion and Outlook") and added new research perspectives for deep, hot biosphere studies at the interface of sediment/hot basalt intrusions. These chemically complex interfaces are underlying the deep Ringvent sediments where we have consistently found high-temperature archaeal communities, and might constitute a new deep subsurface ecological niche (lines 451-461).

With regard to text, the headings and introductions to Results & Discussion sections have been revised, to emphasize the nature of the most significant findings and their context: "Metagenomic features with wide subsurface distribution", line 245 [formerly "metagenomic adaptations"]; "Characteristics and distribution of dominant bacterial and archaeal groups", line 292, with subsections "Dominant subsurface bacteria" (line 302), "Dominant subsurface archaea" (line 335), and "Hadarchaeotal genomic features" (line 379). Finally, new headers were included for the last sections "Genome size trends in the deep biosphere" (line 401), and "Temperature impact on MAG recovery" (line 422).

Some detailed comments:

Introduction: The introduction on biodiversity, activity and functions of microbial communities of the Guaymas Basin, are not sufficient to allow readers to understand the significance and necessity of this study; and the scientific question and objectives of the study were not clearly stated.

Response: We have now stated our research rationale explicitly, first by introducing the recent detection of microbial cells and activity in the deep, hot biosphere (Beulig et al. 2022; Heuer et al. 2020), and the lack of further identification that makes it impossible to specify the biological diversity and potential of the different kinds of bacteria and archaea in the deep, hot biosphere. Therefore we note that bacterial and archaeal members of the deep, hot biosphere and their activity need to be identified by sequence-based approaches (new text lines 51 to 56). A new opening line for the next paragraph (line 61) establishes the link to Guaymas Basin as a suitable field site for this research task. The scientific goals are now stated more specifically and accurately in lines 98 to 104.

Line 78-79, this sentence is not clear. what exactly do you mean by "extent and characteristics" ?

Response: We have replaced these words by the more specific terms "the spatial extent, diversity and activity" (lines 74-75).

Line 92-93, this seems to be the scientific question the MS wanted to address, however, there were too many questions within this sentence, e.g., environmental distribution, genomic potential, adaptation, stress tolerance. Although each of the above questions was mentioned in the manuscript, none of them were clearly and solidly illustrated. I would suggest the authors to be more focus on one or two points and make in-depth analysis and discussion.

Response: We have re-focused this section on environmental distribution and genomic potential of subsurface microbiota; adaptation and stress tolerance are special aspects of genomic adaptation that will be highlighted at suitable occasions in the manuscript text. We have also inserted an introductory line for this paragraph "These contrasting sites provide an opportunity for a comprehensive analysis of subsurface microbiota" (line 89), to link the drilling sites (described briefly in the previous paragraph) to the scientific questions that are introduced here.

Line 129, what do you mean by "incomplete downcore sulfate depletion"?

Response: We have changed this term to "show gradual downcore sulfate consumption (from 27.9 to 18.8 mM)"; the sulfate concentrations in parentheses make it clear that this is not depletion to zero sulfate. (line 131).

Line 142-143, the authors measured petroleum hydrocarbon, why not involving them in later biostatistical analysis?

Response: The problem is that these data have numerous gaps; the high cost of these analyses prevented analyzing a fully matching suite of samples. Thus, we are reporting these data in Supplementary Table 1 and in Supplemental Figures 1A and 1B, but due to extensive data gaps we cannot include them in the correlational analysis.

Line 148-149, why did negative controls have sequenced reads and matched with MAGs?

Response:

While laboratory extraction (laboratory and kit contamination potential) controls may not produce DNA in quantifiable concentrations (as for our two blank controls), they nearly always produce sequence data, and these same data can show up in the data sets for samples prepared with those kits, particularly when there is very little biomass in the original samples (the "kitome", Salter et al. 2014). Therefore, taxa detected in negative controls must be removed from sample data. We have three

negative controls that produced sequence data, two blank controls (only lab reagents for datasets produced by the Teske and Edgcomb labs using the same kits) and a drilling fluid control (which is surface seawater plus drilling lubricant). The reads recovered for each of the three controls before and after QC are reported in Supplementary Table 2. Originally, contaminants were removed based on taxonomic identity in contamination controls and commonly retrieved suspect taxa in the literature (Salter et al. 2014). As requested, to confirm accurate contamination removal, we mapped control reads to our metagenome assembly and removed all contigs that received mapping with minimum 98% identity over minimum 75% of the read length. Contaminant MAGs were identified in this way (Supplementary Table 3), and they were consistent with published contamination profiles (Salter et al. 2014). This did not remove any of the MAGs discussed in this paper (see lines 524-528, and Supplementary Text paragraph "Accounting for Seawater and Laboratory Contamination").

Line 194-213, As one of the major conclusions in the manuscript, the influence of environmental factors on MAG composition is largely descriptive and lacking of sufficient biostatistical supporting. For example, Line 203-206, the authors claimed different environmental parameters that affected the MAG composition in different temperature zone. However, the results were mainly based on the descriptions of nMDS plot, which was lacking of direct statistical support. Also, the authors only described the parameters with positive correlation with the MAG composition and ignored those with negative correlations. In addition, the validity of using nMDS analysis to study the influence of environmental parameters need to be evaluated. Why not using CCA, RDA, etc. ?

Response: To comment on the assumption that the nMDS plot lacks direct statistical support, it is important to understand that the plot shows the vectors of environmental and biogeochemical parameters that correlate with MAGs after passing a statistical threshold filter. The environmental factors plotted and shown in the nMDS plot have a p value of less than 0.05 based on permutation tests using the environmental factors and MAG abundance data sets.

However, we agree that running a CCA would also be informative since the CCA approach allows to take into consideration non-linear regression between different parameters, which nMDS does not do. CCA results underscored most of the earlier nMDS results, since (in addition to the central importance of temperature) central geochemical parameters (among others, total sulfide, TOC, TN, pH, salinity) are shown to impact MAG composition for distinct thermal regimes. We are now commenting on both analyses (nMDS and CCA) in the main manuscript (see extended text in lines 217-243), and we include the CCA plot as supplementary Figure 5.

Line 215-246, the "Metagenomic subsurface adaptation". (1) What are environmental features of studied sites that are special and distinct from other marine habitats? How does the annotated functions provide adaptation advantage for the MAGs, associated with the special environmental conditions of the studied sites? The linkage between the genomic features and specific environmental condition of the study sites is weak. (2) The identified "adaptation features" seemed to be commonly present in other marine habitats such as marine sediments, so what are the special adaptation mechanisms for the thermally heated subsurface microorganisms?

Response: We are discussing widespread genomic features that were found in GB subsurface microbes, and appear to be maintained in this environment; their persistent occurrence indicates that they are useful and relevant (genes involved in the regulation of sporulation, for example). We do not claim that these features are unique for the Guaymas Basin subsurface, and we are reframing this paragraph to clarify expectations (see lines 245-250). The new title "Metagenomic features with wide subsurface distribution" more accurately describes the results that are presented here. Most importantly, we imply that a sizeable portion of these widely distributed and preserved genes does in fact have adaptive value, and confers an advantage to microbes that contain and maintain them. However, other widely

distributed genes require more subtle interpretations and we weigh the pros and cons of adaptive explanations while providing perspective on the (slow) pace of subsurface evolution (see especially lines 281-290).

Line 239-240, how common did the related functions in the MAGs (percentage)?

Response note: We have now included percentage numbers for these functions, and integrated them into the extensively revised text of this section (lines 261-273).

Line 273-276, this sentence reads quite strange, please modify.

Response: The entire section and the preceding paragraph were rewritten, the sentence is now subdivided as reads as follows (line 341-344): "Archaeal MAGs were dominated by the Thermoproteota, an archaeal phylum consisting of four major lineages, the Thaumarchaeota, Aigarchaeota, Korarchaeota and Bathyarchaeia (Oren & Garrity, 2021). All 11 Thermoproteota MAGs from the Guaymas subsurface belonged to the class Bathyarchaeia; these MAGs were detected at all examined sites..."

Line 358 and other locations, what do you mean by "unique MAGs"? A definition is needed.

Response: "Unique" means that MAGs were dereplicated before further analysis (see new methods text, lines 513-523). After dereplication, the 89 resulting MAGs were distinct from each other, and none of these MAGs were redundant. After this clarification, there is no need for the term "unique" and it was omitted.

Line 357-367, the relationship between MAGs recovery and temperature may need to be revisited by considering the sequencing depth of the metagenomes. As the variation on sequencing depth would greatly impact the MAG recovery rate and reads recruitment. For example, if the metagenomes from the hot regions were generally with less sequencing depth (smaller data size), the observed trends of decreasing MAG recovery with temperature would be questionable.

Response: We now present as Supplementary Table 2 the statistics on sequencing depth for all samples. This shows the number of raw reads obtained per sample, the number of reads post QC, the number of assembled contigs per sample, and the number or contigs retained after removing possible contaminants. We ran ANOVA on the fastQ file sizes/metagenome sequencing depths, and we found no significant differences in metagenome sequencing depth between any of the temperature groups. We don't think that data interpretation is significantly impacted by differences in data recovery from different depths/temperature regimes.

Line 424-425, More details in library construction are needed. Were they whole genome amplified before sequencing? What were the concentration and total amount of DNA extracted from the samples? What is the sequencing depth? How many Gb of data for each metagenome?

Response: We are adding more methods detail (lines 621-625) in section "RNA extraction, library preparation, sequencing, and mapping of RNA reads to the MAGs": "Amplified cDNAs from the DNA-free RNA extracts were prepared using the Ovation RNA-Seq System V2 (Tecan) following manufacturer's suggestions. All steps through cDNA preparation were completed the same day to avoid freeze/thaw cycles." cDNAs were submitted to the Georgia Genomics and Bioinformatics Core for library preparation and sequencing using NextSeq 500 PE 150 High Output (Illumina)."

No whole genome amplification was performed at any point in our study, since the consequences of this amplification step for metagenome composition are just too unpredictable. DNA concentrations are now tabulated in new supplementary table 2 that is also reporting other basic parameters relating to mRNA recovery, and sequencing depth in reads per library.

Line 445-449, Why were there contaminants ? and how could they have been sequenced and with so many reads?

Response: We assume that this comment refers to seawater, drilling and laboratory contamination, not the contamination that is methodologically inherent in the metagenomic reconstruction of MAGs from highly complex natural microbial communities. Deep-sea drilling employs a mixture of seawater and lubricant mud during drilling operations, which carries surface seawater-derived microbial contaminants into the sediment samples. Sample processing on the ship includes steps to remove the surfaces of recovered core samples prior to collection for different analyses. This minimizes the chances of contamination by drilling fluids. Contamination monitoring (using polyfluorinated chemical tracers) was run throughout the drilling operations (see Lever et al. 2006 *Geomicrobiology Journal* 23:517-530 for details) but it is never perfect. Thus, the mixed microbial community in seawater and drilling lubricant (which can be substantial!) needs to be accounted for by a drilling fluid control.

The second type of contaminants are introduced through handling, DNA extraction kits and lab chemicals (the “kitome”, see Salter et al. 2014. *BMC Biol.* 12:87). This contaminant community is commonly less diverse than the seawater + drilling fluid control, but needs to be accounted for as well, based on blank extractions (only lab kit and chemicals, no sample) and the relevant literature. To be sure, we have run two blank extraction controls, since DNA was extracted in two labs (using identical kits); at USC (Amend lab) and at WHOI (Edgcomb lab).

This background on contamination control data handling is now included in a text paragraph and references in Supplementary Materials (“Accounting for Seawater and Laboratory Contamination”), and referenced in the main text (lines 524-528).

Line 475, not clear. How could abundance of metagenomic reads been normalized as "transcripts" per million?

Response: In the context of normalizing metagenome reads, the term “transcript” is an unfortunate misnaming, but is currently established in metagenomic analysis and in the relevant bioinformatics literature. “Transcript per million” or TPM is a unit term used in the bioinformatic analysis of both metatranscriptomes and metagenomes; it is unrelated to the process of transcription. We have revised this sentence and the entire paragraph to better describe TPM as a bioinformatics scale parameter (line 553-562). See also our response to a related question from Reviewer 3 about the normalization procedure inherent in CoverM.

Line 536-537, Please provide the concentration and integrity of the RNAs extracted. Also, provide details on cDNA libraries construction. Were the RNA samples with enough concentration and total amount for transcriptome sequencing?

Response: We have added information on RNA and DNA quantity extracted from each sample, and sequencing outcomes for these extracts, to Supplementary Table 2. Regarding the RNA samples, since these samples are extremely difficult to extract RNA from, we elected not to spend any of our RNA for RNA integrity assessments, but to prioritize removal of DNA, and confirmation of removal of all DNA by PCR after DNase treatment. RNA integrity is likely suboptimal due to the additional steps we introduced in our extraction protocols that included ethanol washes of the sediments. However, numerous other extraction methods and variants that we tried failed to yield RNA in any workable amounts, and the methods used here were the only ones that yielded anything.

Figure 4, Description is not clear. Are those data based on composition of MAGs or entire metagenomes from each site? Also, the resolution of the plot was low.

Response: The data in Figure 4 are based on the composition of MAGs. The figure legend (Figure 4) was revised for clarity and brevity as follows:

“The nMDS plot depicts the correlation of Guaymas Basin MAG occurrence with in-situ environmental parameters. The directions of the arrows indicate a positive or negative correlation among the environmental parameters with the ordination axes, for statistically significant ($p < 0.05$) environmental parameters (plot stress: 0.106). Arrow length reflects the strength of correlation between the environmental parameter and MAG occurrence, with longer lines indicating stronger correlations. The samples are color-coded by site, and their temperature regimes are indicated by shape (circles for 0-20°C; triangles for 20-45°C and squares for 45-60°C).”

We are providing new figures (nMDS and CCA versions) in the revision, at high resolution.

We have modified the figure legend in the CCA version (Supplementary Figure 5).

The methodologies for biogeochemical parameter measurement seemed to be missing?

Response: The biogeochemical methods are outlined in the Methods chapter of the IODP 385 Expedition report, now included in the reference list as Teske et al. 2021a (lines 474-475).

Reviewer #2 (Remarks to the Author):

In the manuscript “Metagenomic Profiles of Archaea and Bacteria within Thermal and Geochemical Gradients of the Guaymas Basin Deep Subsurface” Geller-McGrath and Mara et al present a genomic resolution of microbes and their transcripts along a temperature gradient in the Guaymas Basin. The authors combine geochemical measurements with genome-resolved metagenomics and transcriptomics. While I think this paper is of merit and it definitely has the potential to be appear in Nature Communications, I do have some reservations regarding the presentation of the results:

1. The manuscript is very technical. While the results are outstanding and interesting per se, I’m missing an interpretation that makes these findings important, e.g. ecosystem functioning. Even the title is very descriptive. So, the story, the meaning of the results needs to be developed.

Response: Thank you for pointing out the need to translate results per se (the technical aspect) into meaningful statements on ecosystem function. Reviewer 1 made a similar comment that the significance was not well elaborated. To emphasize the relevance of our results for ecosystem functioning, we have amended our manuscript as follows:

We elaborate the paragraph on subsurface biogeochemical zonation and add context on the observed trends in major biogeochemical parameters, as well as site-specific observations (lines 128-162); Table 1 was checked and expanded to include TN values. The revised paragraph sets the stage for MAG occurrence patterns and MAG-based ecosystem inferences in the Guaymas subsurface.

We are adding an expanded transcript section, to improve our presentation of MAG expression within the subsurface ecosystem (line numbers 200-215). Interestingly, MAGs that recruit transcriptomes at warm and hot temperatures often belong to order-level lineages from warm sulfur springs, sulfidic warm water, hydrothermal chimneys and sediments.

We re-emphasize our observation that at moderate and warm temperatures the microbial community reacts to chemistry (meaning, it is subject to ecological interactions), whereas in hot sediments chemical factors fade away and MAGs relate to temperature as primary factor (lines 217-243).

2. The headers of the results/discussion section should present a major finding rather than be technical and – in the worst case – state a method or simply a phylum name like “Chloroflexota”, which doesn’t tell the reader anything.

Response: The phylum names are replaced with more general and informative headings, “dominant subsurface bacteria” or “dominant subsurface archaea”, etc. These sections have now more informative introductions that emphasize the ecophysiological relevance of these dominant bacterial and archaeal groups. The introductory lines (now 292-301) are extensively rewritten, followed by suitably modified text in lines 302ff. The section on subsurface Chloroflexota was fleshed out with a new paragraph on their metabolic potential (lines 317-334). Interestingly, MAGs of the dominant bacterial and archaeal phyla in warm and hot sediments are related to order-level lineages that were previously obtained from warm, sulfidic, or hydrothermal environments (lines 200-217), not cold benthic sediments.

We have also replaced the technical headings of sections on MAG genome size and thermal MAG diversity limits with more informative headings (“Genome size trends in the deep biosphere” [line 401] and “Temperature impact on MAG recovery” [line 422]).

3. Although I do like the correlation of MAG recovery/size to temperature or depth, I’d love to see how the functions in the ecosystem are maintained in the metagenomes (not only binned genomes) and metatranscriptomes (mapped to entire (assembled) metagenomes).

Response: A full presentation of functional and metabolic analyses of these metagenomes is outside of the scope of this paper, especially since a graduate student in another lab who has joined us on the Guaymas drilling expedition is working on such an analysis. However, we have followed the reviewer’s

suggestion and have performed a full metagenome analysis, and we now include a summary figure of KEGG categories for the interested reader as Supplementary Figure S4 (with matching Supplementary Table S9). Many functional categories are well maintained, even in the deepest samples for which we could obtain metagenomes. We note the similarity of this overall KEGG category figure to the Chloroflexota version (Supplementary Figure S3), a possible consequence of high Chloroflexota representation in the subsurface microbiome (lines 331-334).

4. How do the results of the current study compare to “Rapid metabolism fosters microbial survival in the deep, hot seafloor biosphere” by Beulig et al. Nature Communications volume 13, Article number: 312 (2022)? I’m missing this comparison because it seems that the current study contradicts the findings of Beulig et al.

Response: The metagenome profiles in this study (dominated by Hadarachaeota at the high temperature end) approach in-situ temperatures of approx. 60C, whereas Beulig et al. 2022, and the related study by Heuer et al. 2020 report microbial activities (without sequence data) at extreme temperatures up to 120C. The difference is easily understood by the obligatory DNA demands for metagenomic analysis; generally at least 0.5 ng good-quality DNA is required for metagenomic library construction (while avoiding whole genome amplification due to its uncontrolled biases). The very small cell numbers that were observed by Heuer and Beulig in extremely hot subsurface sediments (occasional peaks of ca. 100 cells/cm³ sediment) are not sufficient for metagenomic detection. Without belaboring the point, we are referring to this issue (lines 53 to 56). We are currently finalizing a separate methods manuscript explicitly on this topic.

Minor comments:

L 36 and elsewhere in the manuscript: It should be “By contrast, xxx” or “In contrast to XY, xxx” but not “In contrast, xxx”.

Response: Corrected as suggested (line 36).

L 95: consider rephrasing “mRNA evidence”.

Response: what about this rephrased sentence “...We also provide evidence for the activity of specific bacterial and archaeal lineages by mRNA transcript mapping on bacterial and archaeal MAGs.” (lines 93-95)

L199-203: Have the authors considered using a bioENV to figure out the best combination of environmental variables that described the community composition?

Response: We considered bioENV but did not elect this type of analysis that is based on Euclidean distances. Instead, we used different multivariate analyses (direct and indirect) as per Reviewer 1’s request. We believe that multivariate analyses such as Canonical Correspondence Analysis (CCA) that account for compositional variances are appropriate.

L336: Have the authors considered genome completeness of the MAGs? Taking the completeness into account when it comes genome sizes might be important but a metric other than CheckM needs to be used, for which markers are equally distributed across the genome or generally takes the unequal distribution into account.

Response: We used CheckM2 to estimate completeness, as indicated in the methods. We agree that CheckM has its limitations because the marker genes it looks for are not equally distributed across the genome. CheckM2, by comparison to CheckM, uses machine learning models trained on genome properties including genome length, number of coding sequences, individual amino acid counts, as well as annotation of predicted proteins using KEGG (in the context of KEGG modules and pathways) (Chklovski et al. 2023). A literature search for alternate methods confirmed that CheckM2 is currently

the gold standard for genome completeness assessments, acknowledged even by authors who are introducing alternate bioinformatics tools (often fine-tuned for specialized microbial groups).

L369: How can DNA/RNA recovery overcome cell density? I do not understand the ecological context here.

Response: We wanted to say that improved DNA/RNA recovery can compensate the limitations of low cell numbers (and thus, DNA/RNA content). “While improved DNA and RNA recovery could potentially compensate for declining downcore cell density, ...”(line 434)

L431: Have the authors considered combined assemblies of metagenomes and metatranscriptomes to promote recruitment of reads? I’m not saying this is a necessity, but it could improve the mapping frequency. What is the rationale behind co-assembling all the 26 metagenomes rather than individually assembling the metagenomes?

Response: This is an interesting question; there is virtually no limit to the possibilities of different types of analyses one could do with such data. We did not consider a co-assembly of transcriptomes and metagenomes in this study, but this could be attempted separately, as we believe adding additional analyses would overburden our current paper that is already “criticized” for being too complex. Regarding the metagenomes, 16S rRNA gene profiles (bacterial data not shown in this paper, however archaeal 16S data are presented in Mara et al. 2023 in press, The ISME J) indicate a gradient of community composition with depth (as opposed to distinctly separate communities in every sample), thus we suspected that metagenomes would similarly exhibit a gradient of overlap in composition. Our results support this. Given the complexity of sediment communities we felt that a co-assembly would improve the yield of MAGs. Higher metagenome sequencing depth obtained by co-assembly can yield a more robust assembly that captures more of the diversity in the system, and it facilitates the comparison across samples by giving us one reference assembly to use. It can also improve our ability to recover genomes from metagenomes through differential coverage.

Figure 1: In-situ should be italics (in figure and in legend)

Response: done as suggested.

Figure 6: MAG recovery can be affected by many biological features (e.g., strain heterogeneity). I think a different metric for diversity should be used, which is based on marker genes like ribosomal protein S3 to identify if temperature and depth have an impact on the recovery of genomes / diversity per se.

Response: We agree that MAG diversity should never be interpreted as portraying the complete picture of phylogenetic diversity, and we also never claim that our MAGs represent total diversity. The objective of analysis of assembled MAGs in this study is to learn more about the genomic potential of the taxa whose MAGs are recovered. A separate paper on marker gene profiling of the bacterial and archaeal community is currently being drafted in our group.

Reviewer #3 (Remarks to the Author):

In “Metagenomic profiles of archaea and bacteria within thermal and geochemical gradients of the Guaymas Basin deep subsurface,” the authors examine trends in microbial community composition and metabolic potential by examining several MAGs recovered from cores at various points in Guaymas Basin.

Overall, the manuscript is well-written and the methods are sound. My primary piece of feedback is more of a question: why focus only on MAGs for this study rather than also include results from the full metagenomes? As the authors note, there are limitations and perhaps biases to MAG recovery that result from challenges like poor assembly, poor read recruitment, etc, and it is possible that these limitations may introduce biases with depth, skewing the results.

It seems to me that including some results from an analysis of the full metagenome (i.e. a broader survey of community composition using something like mmseqs2, and/or a broader survey of total metabolic potential focusing on relative abundance of key metabolic genes independent of MAGs) would provide broader context for how the whole community changes with depth independent of the MAGs, and would ameliorate potential biases in MAG recovery that may occur with depth.

Response: Thank you for your positive and thoughtful evaluation of our efforts, and your suggestions to develop this study further. Examining the total metabolic potential of the subsurface community in greater depth is certainly a possible research direction, but as indicated above in a response to another reviewer, we have not taken this path for two reasons: 1) a total metagenomic survey is already being done by a graduate student who sailed on IODP expedition 385 (although on a somewhat different sample set), and we have deliberately coordinated our manuscripts to avoid overlap. This division of labor goes back to the Guaymas drilling cruise in 2019 when this was agreed upon. 2) In this paper, we really want to emphasize the phylogenomic dimension, and examine how microorganisms and their genomic capabilities change downcore; necessarily this means working with MAGs. On their own, core metabolic genes are affected too much by lateral gene transfer.

Nonetheless, also in response to reviewer 2, we have added a KEGG category re-analysis of the metagenomes for all our samples, and we present the resulting overview figure of KEGG functional gene categories for the interested reader (Chloroflexota metagenome in Supplemental Figure 3, and complete microbial community metagenome in Supplemental Figure 4), to provide an idea of the overall functional potential and its downcore changes within our sample set (lines 317-334). Tables of the KEGG abundance results is also added to the supplementals (Supplementary Tables 8 and 9).

In addition to this, the authors put a lot of information in the supplement, but some parts of the Results/Discussion feel overly broad and/or incomplete. I would suggest adding some details back into the Results/Discussion, especially as it pertains to specific metabolic potential of specific taxa/bins and how that changes with depth. As a further example, Supplementary Figure 2 appears to simply be a copy of Figure 2, with a bit more taxonomic detail. It seems to me that these details should be included in the main Figure 2 and there doesn't need to be an additional figure in the supplement.

Response: In writing this manuscript, we have explored numerous possibilities, starting with minimal supplements and a very lengthy manuscript, then turning around to a rigorously pruned manuscript, and finally crafting a compromise where significant text sections were re-integrated into the manuscript (the submitted version). It is certainly possible to re-insert further supplemental materials into the manuscript, yet that another reviewer was unhappy about the relative abundance of descriptive material on various microbial lineages. To solve this dilemma within the context of a manuscript that was growing considerably during revision (ca. 1000 words), we have constructed more informative linkages between the main manuscript and the supplements (beginning with lines 178-188). In the end, we hope to have crafted a compromise that is acceptable for readers who love microbial detail, and for

those who tend to skip this over.

About Figure 2, we are inserting the fully labeled version into the manuscript, and remove the supplemental version.

Further comments are listed below.

Introduction

Line 48: maybe clarify that this would be downcore in cool/moderate temps, right? These are subsurface sediments away from hydrothermal activity?

Response: We have modified the sentence as follows: "As microbial communities in cool, relatively shallow subsurface sediments transition into more deeply buried and increasingly warm and finally hot sediments, it should be possible to track how subsurface bacteria and archaea react to these gradually harsher regimes downcore on the levels of cellular activity and community change." (lines 44-48)
Yes, the sediments are not hydrothermal sediments in the usual sense (poorly consolidated sediments that are permeated by pulsating hydrothermal fluids); they are consolidated subsurface sediments, heated by the strong geothermal heatflow that is pervasive in Guaymas Basin (Neumann et al. 2023, in ref. list).

Methods:

Line 435 What read trimming technique was used?

Response: thank you for pointing out this methods gap. We have added the following sentence and references to the methods: "Before assembly, reads were trimmed for quality and adapters removed using Trimmomatic v0.39 (Bolger et al. 2014) (parameters: leading:20; trailing:20; sliding window: 0-24; min length 50). The quality of reads was verified with FastQC v0.11.9 (Andrews et al. 2012)." (lines 505-508).

References: Bolger AM, Lohse M, Usadel B. Trimmomatic: a flexible trimmer for Illumina sequence data. *Bioinformatics*. 2014 Aug 1;30(15):2114–20.

Andrews S, Krueger F, Segonds-Pichon A, Biggins L, Krueger C, Wingett S. 2012. FastQC: a quality control tool for high throughput sequence data. Babraham Institute, UK.

[:http://www.bioinformatics.babraham.ac.uk/projects/fastqc](http://www.bioinformatics.babraham.ac.uk/projects/fastqc).

This is just a suggestion, but if the authors are trying to reconstruct metabolic modules and the pipelines they use can't do archaea, perhaps try KEGGdecoder, which is more agnostic.

For the most part, though, the methods make sense and it seems that appropriate controls were run and appropriate precautions were taken regarding contamination.

Response: Thank you for the thoughtful suggestion, it would be a natural follow-up to our metagenomic analyses. However, if we reconstructed metabolic modules and emphasized them in our paper, we would infringe on the research (and the manuscript) of a graduate student who sailed with us, and potentially cause problems at a very vulnerable career stage.

However, the implicit suggestion to at least sketch out a physiological downcore profile makes good sense, and we include an overview of functional gene potential within all samples based on KEGG categories as new Supplementary Figure 4, and a corresponding overview of the dominant bacterial phylum Chloroflexota as Supplementary Figure 3. The KEGG abundance data underlying these supplementary heatmaps are also provided as Supplementary tables 8 and 9.

Results and Discussion:

Line 116: remove "at": "cell counts decreased by four to five orders of magnitude"

Response: good catch, corrected.

For the section on Subsurface biogeochemical zonation: a lot of chemistry data is reported in table 1. This is a bit hard to visualize in context of samples—would it be possible to show this as a figure somehow?

Response: Conceptually, such a figure would combine data from different sampling sites and different gradients, in very rough resolution of tens of meters. Individual data points taken out of site-specific context (see figures in IODP site chapters) would be hard to follow. Each datapoint from a chemical profile fits ultimately into the context of its own site and chemical species, as plotted in the site chapters (which we reference).

The contextualization of these environmental data is also presented in the nMDS plot (Figure 4) and in the CCA plot (Supplementary Figure 5). In fact, the purpose of these plots is to extract from the table the factors that are driving MAG distribution.

Yet, to help to track the chemistry in the context of the samples, we have rearranged Table 1 to show each site in downcore sequence.

It is unclear to me what units are in the x-axis of supplementary figure 1.

Response: Good catch, this is temperature in Celsius (the C is cut off by mistake). We are providing a corrected figure, now with panels 1A and 1B for total petroleum hydrocarbons and saturated hydrocarbons, respectively.

For the section on MAG diversity, distribution and evidence of activity:

The authors comment on relative abundances of MAGs in different depths, which is interesting, but why are specific MAGs within a specific phylum more abundant than others? More description of differences between specific MAGs (sort of like a mini pangenome analysis) would help contextualize these results. For example, why are some Chloroflexi at high abundance and not others? Is there information that might contextualize this based on the metabolic pathway completeness analyses?

Response: This is an excellent comment, and this kind of comparison of genome content for presumably abundant taxa vs rarer taxa would frame a potential follow-up study. For this study, we wanted to dig into this a bit for the Chloroflexota, which appear to be one of the most successful and diverse groups present in our MAGs and in the marine sedimentary subsurface in general (Fincker et al. 2020). We have created a new KEGG category figure that shows KEGG category abundances for all Chloroflexota in a heatmap format (Supplementary Figure 3, and matching abundance table (Supplementary table 8) (text lines 317-334). The figure shows that the Chloroflexota are remarkably consistent in their downcore metagenomic profiles across different samples, and are gradually reduced only at ca. 40C and higher. If the editor wishes to approve an additional main text figure, it can be moved into the main text.

Along the lines of the above comment, a figure showing bin metabolic pathway completeness heatmap to compare with figures 2 and 3 would be helpful. (KEGGDecoder makes heatmaps like these, but I am sure other pipelines do too.)

Response: We are providing the underlying KEGG table for Chloroflexi MAGs (in excel format) as Supplemental Table 8, which matches the heatmap figure (Supplementary Figure 3). This large table provides access to the abundance of relevant Chloroflexi pathways and gene categories from the Guaymas subsurface, a useful service for the (considerable) group of Chloroflexi researchers who might want to dig into these data further.

Lines 171-178: The section on metabolisms, and the mention of CRISPRs, needs to be contextualized, as it is seemingly thrown in there with no further discussion. What can we derive from this?

Response: This information is included and discussed in more detail in the supplementary information, which is now referenced explicitly in expanded lines of the manuscript text (lines 179-188). Mentions of CRISPR detection are now limited to the supplement, where more context is available.

Figure 2: For relative abundance, did the authors normalize according to the length of the MAG? The values are reported as what percent of reads per sample mapped to a MAG, so it's normalized for the number of reads per sample, but in addition to that it seems like MAGs that are longer would have more reads that recruit to them. I checked Materials and Methods and this isn't clear – is this normalization calculated as part of CoverM?

Response: This question (and a related comment from Reviewer 1 about normalization units) prompted us to check the inner workings of normalization in CoverM with the developer. It turned out that CoverM's "relative abundance" metric (that we used when mapping metagenomic and metatranscriptomic reads to our MAGs) accounts for genome size by using the mean coverage of the genomes (as opposed to the total coverage). For example, if we had two MAGs that were equally abundant, but one MAG was 1Mbp in length and the other was 2Mbp, double the reads would map to the MAG that was 2Mbp in length. To account for this, CoverM compares the proportion of mean coverages of the input set of MAGs in its calculation of relative abundance, which would be the same for these two genomes in this example. Its calculation therefore not only accounts for total metagenomic/metatranscriptomic library size, but genome length as well. However, to test for potential effects of different normalization approaches, we also normalized reads mapped to kilobase of genome, per million reads (the rkm normalization option in CoverM, see table of normalization options in <https://github.com/wwood/CoverM#calculation-methods>). Using this alternate normalization, Figures 2 and 3 looked practically indistinguishable for the original versions; thus we keep the current (original) versions, after improving them with taxonomy annotation.

For the section on metatranscriptome mapping (lines 179-194) (now starting at line 200):

As above, this section would benefit from a bit more detail and context. Why are specific MAGs more active than others within specific phyla? Were certain metabolisms more active than others as depth and/or temperature increased? Were these metabolisms specific to certain phyla? How did activity compare with overall DNA abundance?

Response: Prompted by this comment, we re-examined Figure 3, which shows the transcriptionally active MAGs. Interestingly, it turned out that most bacterial and archaeal MAGs that were actively transcribed at warm and hot temperatures belonged to lineages that were previously detected in warm sulfidic springs, in hydrothermal carbonate chimneys, and even in hydrothermal sediments of Guaymas Basin. We recognize that we have definitely missed some opportunities here, and are now providing phylogenetic and environmental context for this section (lines 200-215). For easy sample identification, the samples of Figure 3 are taxonomically annotated (on the y-axis).

In examining the supplementary material, it seems that a lot of this information is in the supplementary material, but without this context in the main manuscript, the results are overly vague. Please consider perhaps condensing much of the info in the supplementary materials and putting it in the main manuscript.

Response: We have added more specific material to the main manuscript: a sharper scientific rationale in the introduction; revised Figures 2 and 3 fully annotated with MAG taxonomy; revised Figure 4 (and matching Supplementary Figure 5) with updated geochemistry correlations; new discussion of transcriptomics in warm and hot samples; and numerous text emendations and revisions throughout the manuscript. With all these additions, we are now running over the 5000-word limit. However, we have inserted more supplement references (text, figure and table links) while eliminating unnecessary

duplications in the main text (see lines 179-188).

Add “that” to line 219: “widespread genomic features that extend”

Response: Good, done

Lines 241-248: this is an interesting result; the authors should have the data indicating whether genes for chemotaxis were actually expressed in the MAGs, no? It seems that would be worth reporting here.

Response: Yes, the chemotaxis genes and motility genes are expressed primarily between 0.8 and 36.8 mbsf, and this is reported in a separate paper in press (Mara et al. ISME J). We have inserted a new paragraph where we highlight the slow pace of evolution in the subsurface, and the potential of microorganisms to retain chemotaxis and motility genes in the subsurface (Lines 272-290).

Supplementary Figure 2 seems like it’s exactly figure 2, but with more detail regarding which taxa each row correspond to. It seems like that should be included in figure 2, and supplementary figure 2 could be dispensed with.

Response: We are reorganizing our figures; Figure 2 is now fully labeled with taxonomy information on the y-axis, and then the supplementary version is no longer needed. Figure 3 is also fully labeled for easier reference.

Consider reorganizing the text: most of the phylum-specific MAG description is focused on abundance (DNA), not RNA. One suggestion would be to add the metatranscriptomic results later, and/or add metatranscriptomic results to phylum-specific descriptions.

Response: In this manuscript, metatranscriptomic results are essentially used to validate DNA and MAG results by transcript mapping, something that was not possible in our separate overview manuscript on subsurface metatranscriptomics in Guaymas Basin (Mara et al. ISME J., in press). We make this connection more explicit so that readers will not miss it (lines 189-191): “To determine any intra-phylum differences in metabolic activity, we mapped reads of the Guaymas Basin subsurface metatranscriptome (Mara et al., in press) to our recovered MAGs, for samples collected at the same sites (Figure 3).” To avoid overlap with our own paper (Mara et al., in press) we limit the transcriptomics results to validate MAG activity, and for this purpose they are introduced within the section “MAG diversity, distribution, and evidence of activity” (lines 200-215).

As I mentioned above, I also think this section would be helped by a metabolic pathway completion map like those produced by KEGGdecoder. Would help visualize differences between these MAGs, visualize metabolic potential.

Response: We would like to keep the phylogenomic gradient (MAG diversity changes downcore) in the center of our manuscript, and avoid overlap with the ongoing metabolism-centered study of our graduate student colleague. However, for overview purposes we are now including a KEGG category analysis of the metagenome samples as Supplemental Figure 4, and its Chloroflexota-focused equivalent as Suppl. Figure 3.

Rika Anderson

Reviewer #1 (Remarks to the Author):

The authors made great effort to modify the manuscript and my concerns on methodologies are largely eliminated. However, some shortcomings still remain (1) The major body of the manuscript is mainly descriptive information, and the results and discussion focused too much on scattered details and lack sufficient integration and in-depth analysis. (2) The significance and impact of the findings were not clearly discussed and illustrated.

The metagenomic and metagenomic data sets from such extreme habitat on Earth are undoubtedly with great values, and the discovery of temperature regulated distribution of microbial communities and preference of different taxa in different temperature are very interesting. However, the manuscript still needs to be substantially re-organized, making good summarization, integration and discussion of all geochemical, distribution, as well as functional features. The hypothesis ' that the Guaymas Basin subsurface community is not a random assemblage, but reveals phylogenetic and functional structure that can be tracked downcore. ' is a good and attractive point. Unfortunately, current version of MS focused mainly on phylogenetic structure while the patterns of functional characteristics along the temperature gradient is not clearly shown. I suggest the authors to enhance the analysis on changes of functional composition at different temperature zone, revealing the possible survival strategy and ecological/biogeochemical effects of microbes at different temperature zone, and if possible, find out the genomic features for microbes to adapt to high temperature. Correspondingly, at least one figure should be included in the main text to show the functional features of the microorganisms. Figures showing similar kind of data can be integrated, for example, Figure 2 and 3 can be integrated into one.

Please note that "enhance the analysis of functional characteristics" not means putting more details of functional annotations in the main text, but to make in-depth and integrative analysis on a particular question such as "changes of functional characteristics along the temperature gradient and their implications".

Would it be better to show the changes of MAG recovery (i.e., MAG diversity) and MAGs composition along the temperature gradient first, then shown the changes of genomic features (Genome size, and functional characteristics composition) to reveal the functional and metabolic basis for observed species patterns? (This is my personal feeling about the logic, just for communication)

Some other comments:

Line 52, why would "sequence-based identification" be so important?

Line 55-57, why is it important to "learn more about bacterial and archaeal communities of the deep, hot biosphere from a genomic perspective"?

The authors provided details of DNA/RNA extractions in table S2. I found that many samples were with low DNA/RNA concentration (especially for the negative controls), how can they be successfully sequenced without amplification (low quantities of nucleic acid usually lead to failure in library construction)? What kit was used for metagenomic library construction? Was it the same kit for library construction for the other samples?

Line 245-399, this part is very long but I feel it can be organized and analyzed in a better way. A lot of content here are repeating the distribution of different species.

Reviewer #2 (Remarks to the Author):

The authors have - though many times referring to other studies that are in preparation / accepted elsewhere - all my questions. The only remaining question that I have relates to Figure 6. The authors perform a correlation analysis with unique MAGs. However, there might be other explanation than temperature and depth for a decline / increase in recovered MAGs. Can the

authors please also test alternate explanations and correlate the number of unique MAGs with sequencing depth (no of sequenced bps per sample) or DNA yield (from DNA extractions per sample)? Lastly a FDR-correction of the p-values retrieved from these correlations would be appropriate.

I congratulate the authors on this great study.

Reviewer #3 (Remarks to the Author):

The authors have sufficiently and thoughtfully addressed my comments, and I feel that the paper is now suitable for publication.

Reviewer responses

Reviewer #1 (Remarks to the Author):

The authors made great effort to modify the manuscript and my concerns on methodologies are largely eliminated. However, some shortcomings still remain (1) The major body of the manuscript is mainly descriptive information, and the results and discussion focused too much on scattered details and lack sufficient integration and in-depth analysis. (2) The significance and impact of the findings were not clearly discussed and illustrated.

Response: We respectfully disagree on both points. The manuscript has been carefully calibrated to strike an acceptable balance between emphasizing the overall scientific message and adding illustrative detail, first during repeated rewrites among the author team, and then during the review process. Colleagues, editors and reviewers clearly understand the general significance and the scientific implications of our manuscript.

The metagenomic and metagenomic data sets from such extreme habitat on Earth are undoubtedly with great values, and the discovery of temperature regulated distribution of microbial communities and preference of different taxa in different temperature are very interesting. However, the manuscript still needs to be substantially re-organized, making good summarization, integration and discussion of all geochemical, distribution, as well as functional features. The hypothesis 'that the Guaymas Basin subsurface community is not a random assemblage, but reveals phylogenetic and functional structure that can be tracked downcore.' is a good and attractive point. Unfortunately, current version of MS focused mainly on phylogenetic structure while the patterns of functional characteristics along the temperature gradient is not clearly shown. I suggest the authors to enhance the analysis on changes of functional composition at different temperature zone, revealing the possible survival strategy and ecological/biogeochemical effects of microbes at different temperature zone, and if possible, find out the genomic features for microbes to adapt to high temperature. Correspondingly, at least one figure should be included in the main text to show the functional features of the microorganisms. Figures showing similar kind of data can be integrated, for example, Figure 2 and 3 can be integrated into one.

Response: The fundamental problem in a paper of high complexity is to choose a particular perspective that maximizes insight and understanding, and we have chosen phylogeny in first place and function in second place. The reason is that phylogenetic composition changes downcore more significantly and more visibly than functional potential. Supplementary figures S3 and 4, the KEGG heat maps of functional gene categories, show most functional categories initially persist and then decline in a similar manner as depth and temperature increase, reminiscent of similar observations in surficial hydrothermal sediments (Su et al., 2023. AEM, 10.1128/aem.00018-23). So, instead of arguing from a functional perspective first and foremost, we anchor our manuscript in MAG phylogenetic diversity trends, and then link functional inference to noteworthy MAGs. Of course, the interested reader will find plenty of

functional information in the text and tables, especially in the supplements. This is a goldmine that is open for further exploration.

Figures 2 and 3 show results for metagenomic and transcriptomic analyses, respectively. Each of these large figures is already near the limit of complexity and readability (for example, regarding taxon labels on the y-axis). Apart from the unsolved question how to “combine” these figures, a combination would produce an overly large figure that does not fit on a page and (if reduction in scale is attempted) becomes impossible to decipher.

Please note that "enhance the analysis of functional characteristics" not means putting more details of functional annotations in the main text, but to make in-depth and integrative analysis on a particular question such as "changes of functional characteristics along the temperature gradient and their implications".

Response: This is again a matter of perspective. In our strategy, the “integrative analysis” focuses on MAG diversity first, and from there it is possible to infer functional diversity. This strategy has the advantage of greater specificity; it is more accurate to argue from the organisms (represented by MAGs, see Figures 2 and 3) towards their potential ecophysiological function, than to start with function and trying to split up widely shared and distributed functionalities (see Supplementary Figures 3 and 4) into numerous and diverse MAGs that are participating.

Would it be better to show the changes of MAG recovery (i.e., MAG diversity) and MAGs composition along the temperature gradient first, then shown the changes of genomic features(Genome size, and functional characteristics composition) to reveal the functional and metabolic basis for observed species patterns? (This is my personal feeling about the logic, just for communication)

Response: We have placed changes in MAG recovery last, in order to highlight the limits of MAG detection at 60C, and to showcase again the crucial role of temperature that delimits the deep subsurface biosphere. Ending this study with a focus on temperature limits also facilitates comparison with other deep biosphere studies (which provide temperature information but usually do not contain information on MAG genome size trends).

Some other comments:

Line 52, why would "sequence-based identification" be so important?

Response: Sequence-based identification provides a phylogenetic and physiological identity to microbial populations that are otherwise only known from cell counts.

Line 55-57, why is it important to "learn more about bacterial and archaeal communities of the deep, hot biosphere from a genomic perspective"?

Response: Since these deep subsurface communities are resistant to cultivation and lab-based study of isolates, genomic analysis is (for now) the best way to study them. The next generations of deep subsurface microbiologists will hopefully have more cultures at their disposal.

The authors provided details of DNA/RNA extractions in table S2. I found that many samples were with low DNA/RNA concentration (especially for the negative controls), how can they be successfully sequenced without amplification (low quantities of nucleic acid usually lead to failure in library construction)? What kit was used for metagenomic library construction? Was it the same kit for library construction for the other samples?

Response: Our sequencing libraries were prepared at the Sequencing center of the University of Delaware, which specializes in successful library construction from low-DNA samples. We are discussing DNA yields and methodological requirements in an upcoming paper that has been submitted to the Results volume of the IODP 385 Expedition reports. The low RNA concentrations were something that we expected because the Guaymas subsurface sediments were hydrocarbon-rich. We constructed the cDNA libraries from total RNA using the Ovation RNA-Seq System V2 kit (NuGEN) which uses picograms of template RNA. As in the case of DNA, the cDNA libraries were sent to a sequencing facility (Georgia Genomics and Bioinformatics Core facility) which also specializes in challenging, low biomass samples.

Line 245-399, this part is very long but I feel it can be organized and analyzed in a better way. A lot of content here are repeating the distribution of different species.

Response: In the re-revised manuscript, text in these lines is subdivided into contrasting sections that are thematically distinct, while progressing logically from shared genomic features of the subsurface community towards more specific content on particular bacteria and archaea: “Metagenomic features with wide subsurface distribution” (lines 236 to 279), “Characteristics and distribution of dominant bacterial and archaeal groups” (line 281 to 289), “Dominant subsurface bacteria” (lines 290 to 321), “Dominant subsurface archaea” (lines 322-363), and “Hadarchaeotal genomic features” (lines 363 to 382). Please note that these sections do not discuss “species” but MAGs, and major taxonomic groups.

Reviewer #2 (Remarks to the Author):

The authors have - though many times referring to other studies that are in preparation / accepted elsewhere - all my questions. The only remaining question that I have relates to Figure 6. The authors perform a correlation analysis with unique MAGs. However, there might be other explanation than temperature and depth for a decline / increase in recovered MAGs. Can the authors please also test alternate explanations and correlate the number of unique MAGs with sequencing depth (no of sequenced bps per sample) or DNA yield (from DNA extractions per sample)? Lastly a FDR-correction of the p-values retrieved from these correlations would be appropriate.

I congratulate the authors on this great study.

Response: Alternative interpretations were tested using the information in Supplementary Table 2, which contains the sequencing depth for all samples (number of raw reads obtained per sample, the number of reads post QC, the number of assembled contigs per sample, and the number of contigs retained after removing possible contaminants). We ran ANOVA on the fastQ file sizes/metagenome sequencing depths, and found no significant differences in metagenome sequencing depth between any of the three temperature groups. Thus, we don't think that data interpretation is significantly impacted by differences in sequence recovery from different depths/temperature regimes.

However, DNA yield is another matter, since declining DNA recovery (a proxy for declining microbial biomass and cell numbers) will intrinsically limit the diversity of MAGs that can be recovered. At some point, DNA "extinction" will stop the recovery of MAGs. This issue is now acknowledged explicitly by rewriting the introductory sentence in the paragraph "Temperature impact on MAG recovery":

"The environmental stresses that increasingly exclude microbial lineages, reduce genome size and reduce overall microbial population size (and thus, quantity of recovered DNA) are reflected in decreased recovery of MAGs in warmer and deeper samples from all sites (Figure 6A, B)." (lines 404-407)

In a recently submitted data report to the Expedition 385 IODP volume, we are revisiting this issue more extensively (some reviewers might think obsessively) by examining declining DNA yield trends with depth and temperature for multiple sample sets. DNA yields (and parallel cell counts) drop dramatically at hot sites (Ringvent) and much more gradually at more temperate sites (all others). We do not rule out that improvements in DNA extraction could at some point yield MAGs from sediments with higher temperatures. As long as cells exist, there should be DNA present; the question is how to recover it in sufficient amounts and quality.

Thank you for your constructive and eminently actionable comments, they have contributed substantially to the revision and improvement of our study.

Reviewer #3 (Remarks to the Author):

The authors have sufficiently and thoughtfully addressed my comments, and I feel that the paper is now suitable for publication.

Response: Thank you for your constructive and specific comments, they have contributed substantially to the revision and improvement of our study.